



# Snow Avalanche Detection and Mapping in single, multitemporal, and multiorbital Radar Images from TerraSAR-X and Sentinel-1

Silvan Leinss[*1], Raphael Wicki[*1], Sämi Holenstein[1], Simone Baffelli[1], and Yves Bühler[2]

[*]These authors contribute equally to the paper
[1]Institute of Environmental Engineering, ETH Zurich, Zurich, Switzerland
[2]WSL-Institute for Snow and Avalanche Research SLF, Davos Dorf, Switzerland

**Correspondence:** Silvan Leinss (leinss@ifu.baug.ethz.ch)

**Abstract.** Snow avalanches can endanger people and infrastructure, especially in densely populated mountainous regions. In Switzerland, the public is informed by an avalanche bulletin issued twice a day during winter which is based on weather information and snow and avalanches reports from a network of observers. During bad weather, however, information about occurred avalanches can be scarce or even be missing completely. To asses the potential of weather independent radar satellites we compared manual and automatic avalanche mapping results from high resolution TerraSAR-X (TSX) stripmap images and from medium resolution Sentinel-1 (S1) interferometric wide swath images. Within a selected test site in the central Swiss Alps the TSX results were also compared to available mapping results from high-resolution SPOT-6 optical satellite images. We found that avalanche outlines from TSX and S1 agree well with each other but with TSX about 40% more, mainly smaller avalanches were detected. However, S1 provides a much higher spatial and temporal coverage and allows for mapping of the entire Alps at least every 6 days with freely available acquisitions. With costly SPOT-6 images the Alps can be even covered in a single day at meter-resolution, at least for clear sky conditions. For the SPOT-6 and TSX mapping results we found a fair agreement but the temporal information from radar change detection allows for a better separation of overlapping avalanches. Still, with radar, mainly the avalanche deposition zone was detected, whereas the release zone was well visible already in SPOT-6 data. With automatic avalanche mapping we detected around 70 % of the manually mapped new avalanches in the same image pair, at least when the number of old avalanches is low. To further improve the radar mapping capabilities, we combined S1 images from multiple orbits and polarizations and obtained a notable enhancement of resolution and speckle reduction such that the obtained mapping results are almost comparable to the single orbit TSX change detection results. In a multiorbital S1 moasic covering entire Switzerland, we detected 7361 new avalanches which occurred during an extreme avalanche period around Jan 4th 2018.





# 1 Introduction

Snow avalanches frequently threaten people and infrastructure in Switzerland and other mountainous countries. Every winter, dozens of people caught in avalanches suffer serious injuries or even die (Techel et al., 2016) and roads and railways have to be closed during periods of high avalanche danger. To inform about the current avalanche danger levels, ranging from 1 (low) to 5 (very high) on the European Avalanche Hazard Scale (Meister, 1995), the WSL-Institute for Snow and Avalanche Research (SLF) publishes an avalanche bulletin twice a day during winter (SLF, 2018d). The bulletin is written by avalanche experts which analyze weather station data, local snow conditions, detailed weather forecast information and avalanche occurrence reported by a network of in-situ observers. Unfortunately, low visibility due to heavy snow fall and valleys and ski resorts closed due to high avalanche activity can lead to incomplete or missing avalanche occurrence information. In such situations, as happened in Switzerland in January 2018 and 2019, avalanches can be mapped manually in optical airborne images (Bühler et al., 2009; Eckerstorfer et al., 2016; Korzeniowska et al., 2017) or satellite images which have to be tasked in rapid mapping mode (Scott, 2009; Lato et al., 2012; Bühler et al., 2019). The resulting avalanche outlines can then be used to update avalanche databases which are of great value for hazard mapping and mitigation measure planning (Rudolf-Miklau et al., 2014). As manual mapping is very time-consuming, attempts have been made to automatize avalanche mapping in optical data (Bühler et al., 2009; Lato et al., 2012; Frauenfelder et al., 2015; Korzeniowska et al., 2017). To provide weather-independent observations the project Alpine Avalanche Forecast service (AAF) evaluated terrestrial and spaceborne radar images (Bühler et al., 2014). They concluded that medium to large avalanche events could be mapped using very high resolution radar satellites but with the drawbacks of limited availability and high costs. Nevertheless, for freely available but medium-resolution Sentinel-1 radar images few but promising avalanche mapping studies exist (Vickers et al., 2016; Eckerstorfer et al., 2017; Wesselink et al., 2017; Abermann et al., 2019).

To evaluate the applicability of high and medium resolution radar images for avalanche detection in the Swiss Alps we compare 10-meter resolution Sentinel-1 radar images, 3-meter resolution TerraSAR-X radar images, and 1.5 meter resolution SPOT-6 optical images with each other and analyze different methods to detect avalanches from single, multitemporal and multiorbital radar images for two extreme avalanche events which occurred in Switzerland in Jan 2018.

# 2 Study area and data

The study area shown in Fig. 1 was determined by the spatial and temporal availability of high resolution radar images from the satellite TerraSAR-X (TSX), operated by the German Aerospace Center (DLR). No systematic coverage is available because TSX acquires data upon request for scientists and private customers (Werninghaus and Buckreuss, 2010). To cover the two extreme avalanche events around Jan 4th and 22nd 2018 (Fig. 2) with images acquired from identical orbits we searched the TSX archive for a sequence images which limited the study area to the Alps of Uri in central Switzerland. The images were acquired in X-band (9.6 GHz) with the standard TSX stripmap mode (SM) at a resolution of $2.3 \times 3.3$ m (rg$\times$az). Acquisitions are listed in Table 1.


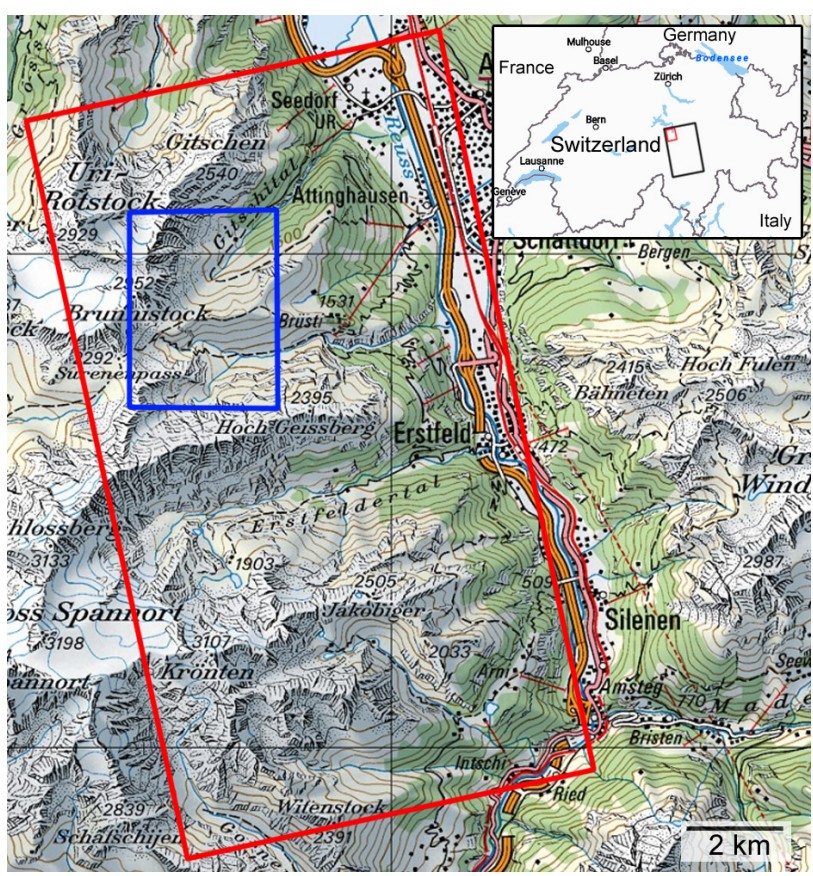

**Figure 1.** Area selected for avalanche mapping (Red rectangle). Blue rectangle: subset used to visualize radar images and mapping results (46°51' N, 8°34' E). The insets shows the footprint of the full TSX scene over Switzerland (black rectangle). © 2019 swisstopo (JD100042), reproduced with the authorisation of swisstopo (JA100120).

**Table 1.** Satellite data with local acquisition time (CET). The TSX and S1 images were acquired looking east from ascending orbits.

| Satellite | Date, local time | mode | pol. | Inc. angle |
|-----------|------------------|------|------|------------|
| TSX | 2017-12-31  18:09 | SM | HH | 29° |
| TSX | 2018-01-11  18:09 | SM | HH | 29° |
| TSX | 2018-02-02  18:09 | SM | HH | 29° |
| S1 | 2017-12-31  18:14 | IW 1 | VV/VH | 34° |
| S1 | 2018-01-12  18:14 | IW 1 | VV/VH | 34° |
| SPOT-6 | 2018-01-24  10:03 | bands: R, G, B, NIR | | 3.1° |

The full TSX scene covers $55{\times}35\,\text{km}^2$ but for the analysis we selected an area of $15.3{\times}8.6\,\text{km}^2$ which shows a very high avalanche activity (red rectangle in Fig. 1). The altitude of the selected area ranges from 400 - 3200 m.a.s.l.. For visualization of results we show in the following only a small subset of the selected area (Fig. 1, blue rectangle).



Radar images of the satellite Sentinel-1 (S1) were analyzed for comparison. S1 images are acquired globally and systematically and are free and openly available for download within 24 hours after acquisition (Davidson et al., 2010). Currently, S1 consists of two satellites, S1-A and S1-B, which alternately image central Europe every six days from the same orbit with a resolution of $2.7 \times 22.5$ m (rg×az) with the interferometric wide swath mode (IW). The S1 images, covering $250 \times 170$ km$^2$,

were selected such that they had orbits and acquisition times similar to TSX. The first analyzed images of both satellites were acquired on 2017-12-31 a few minutes after 18 h local time (Table 1). The second TSX image was acquired on 2018-01-11, one day before the second S1 image (2018-01-12). On the day in between, the avalanche activity was very low (Fig. 2) and the avalanche danger level was moderate for the selected area. Meteorological conditions were relatively stable.

To analyze the second avalanche event, the SLF ordered optical SPOT-6 images which were acquired with the single-pass

multi-strip collection mode through which the most of the Swiss alps ($300 \times 40$ km$^2$) could be imaged in a single day (2018-01-24) at a resolution of 1.5 m. These images were visually searched for avalanches by an expert (Bühler et al., 2019). For comparison we also analyzed TSX data from 2018-02-02, acquired 9 days later.

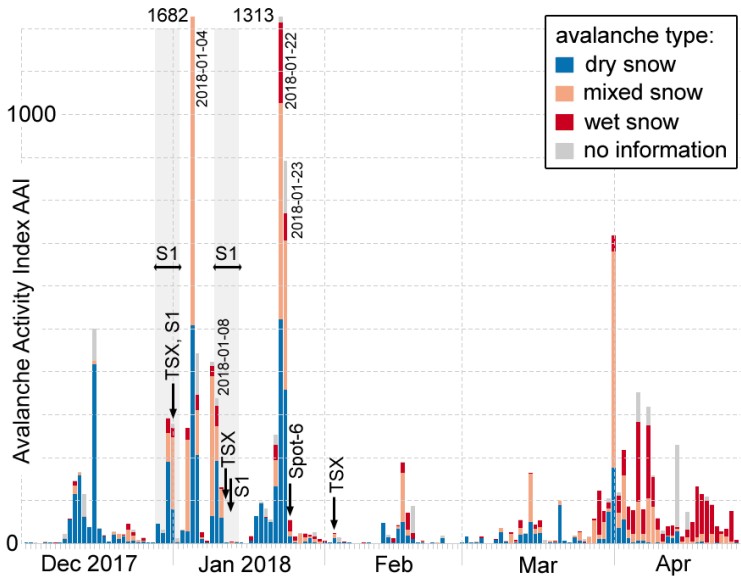

**Figure 2.** Avalanche Activity Index for Switzerland modified after Winkler et al. (2019). Satellite acquisitions dates are indicated by arrows. Images for the multiorbital S1 composite were acquired during the gray shaded periods.

## 2.1 Avalanche events and meteorological conditions

January 2018 was exceptionally warm, humid and stormy. It was the warmest January recorded by systematic measurements

since 1864 and many stations reported record sums for new snow and precipitation (MeteoSchweiz, 2018).

On Jan 3$^{rd}$, embedded in warm and humid winds from the Atlantic Ocean, with a snow line above 1000 m.a.s.l., the storm "Burglind" hit Switzerland and wind speeds up to 200 km h$^{-1}$ were measured on alpine summits. Humid Mediterranean cur-





rents followed the storm. During the first avalanche period from Jan 3–10th, the avalanche danger level was generally 3 (con-

siderable) but raised to 4 (high) on Jan 4/5th, and on the 9th for a major part of the Swiss alps (SLF, 2018a). During sunny

days from Jan 13–15th the snow line raised up to 2200 m.a.s.l. followed by almost daily precipitation and strong winds from

north-west on Jan 16–22nd. Wind speeds of over 100 km h$^{-1}$ occurred during the storms "Evi" and "Friedericke" on Jan 17th

and 18th. On Jan 20/21st extreme snowfall was registered. From Jan 21–23rd, the avalanche activity reached a new three-day

record for the past 19 years since the avalanche winter in 1999 (Winkler et al., 2019). Due to the extraordinary avalanche

situation, the avalanche danger level was raised to 5 (very high) on Jan 21/22nd (SLF, 2018b). After Jan 24th the situation eased

and temperatures were very warm with a snowline rising from 1000 m to over 2500 m until end of January. The avalanche

activity was low and the avalanche danger level was mainly moderate for the analyzed area. On Feb 1st temperature dropped

and around 20 cm of snow fell (SLF, 2018c). Therefore, snow conditions were slightly different between the TSX and SPOT-6

data but only a few new avalanches occurred (Fig. 2).

## 3   Radar backscatter physics of avalanches

We detected avalanches based on the radar backscatter signal and their visual appearance (shape). As illustrated in Fig. 3, all

types of snow avalanches are composed by three different zones (International Commission of Snow and Ice, 1981). The most

upslope part is the release area (blue) with a smooth surface caused by the failure of the weak layer, followed by the zone

of transition (purple) with the stauchwall and some deposition caused by the terrain roughness, and finally the tongue-shaped

zone of deposition (red) at the bottom which is covered by densely compacted snow granules.

Based on snow properties, the different zones show a different radar backscatter signal. In first order scattering physics the

total backscatter intensity of a snow pack, $\sigma^0_{\mathrm{snow}}$, can be composed by scattering from the snow surface, $\sigma^0_{\mathrm{surf}}$, from the snow

volume, $\sigma^0_{\mathrm{vol}}$, from the ground below the snow pack, $\sigma^0_{\mathrm{ground}}$, and from higher order interactions between different structures in

the snow pack $\sigma^0_{\mathrm{inter}}$. Generally, scattering depends on the surface- and interface roughness and on the incidence angle $\theta$.

$$\sigma^0_{\mathrm{snow}}(\theta) = \sigma^0_{\mathrm{surf}}(\theta) + \sigma^0_{\mathrm{vol}}(\theta) + \sigma^0_{\mathrm{ground}}(\theta) + \sigma^0_{\mathrm{inter.}}(\theta) \tag{1}$$

For plain dry snow of few meters depth scattering at the ground usually dominates the signal because microwaves are weakly

scattered at the snow surface and within the volume and penetrate therefore the snow pack to the ground. There, the ground

roughness determines the backscatter signal but for smooth ground mainly forward scattering occurs. For deeper snow volume

scattering can dominate the signal.

In contrast to plain dry snow, snow is deeper and denser in the deposition zone and the surface is rougher. Due to the higher

permittivity, the contribution of $\sigma^0_{\mathrm{vol}}$ and $\sigma^0_{\mathrm{surf}}$ to the total backscatter intensity increases. Both, the rough surface and the volume

scatters radiation more omnidirectional compared to an undisturbed snow pack over smooth ground.

For plain wet snow, however, the incoming radar waves are weakly scattered back at the air-snow interface whereas most

radiation is lost by absorption and forward scattering.

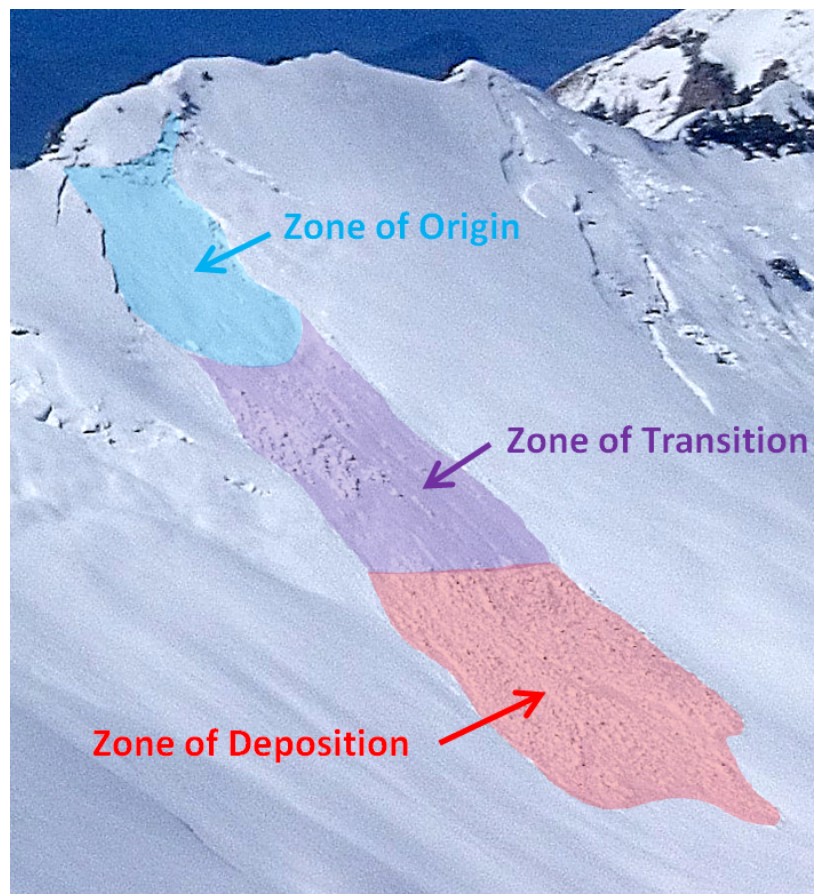

**Figure 3.** Different avalanche zones illustrated by a slab avalanche.

As the volume and ground contribution is negligible for wet snow avalanche debris, the dominant backscatter signal results
from omnidirectional scattering at the increased surface roughness in the deposition zone of avalanches.

In radar images, the zone of origin is very difficult to detect because only the weakly scattering snow volume is reduced
without major changes in the surface roughness. The zone of transition is only sometimes visible, depending on the deposition
of avalanche debris. Therefore, mostly the deposition zone can be detected by a brighter backscatter signal and the mostly
elongated, tongue shaped geometry.

To obtain a high backscatter contrast with respect to the avalanche surrounding the local incidence angle $\theta$ should be far
away from zero to avoid specular backreflection from smooth surfaces. Therefore, the visibility of avalanches in radar images
should be much better for slopes facing off the radar. These slopes are anyway imaged with a higher resolution $\delta_{sr}/\cos\theta$ close
to the slant-range resolution $\delta_{sr}$.


## 4 Methods

### 4.1 Data preprocessing

All radar products were downloaded in the single look complex (SLC) format. The data were preprocessed with the ESA SNAP
Sentinel-1 toolbox and also with the GAMMA software for comparison. The workflow using GAMMA was implemented with
Nextflow (Di Tommaso et al., 2017) to speed up execution and code development and to ensure a reproducible analysis. Preprocessing consists of coregistration, multilooking for reduction of radar speckle (TSX: 6×5 px, S1: 4×1 px), orthorectification,
and generation of radar shadow and layover masks. The SNAP workflow for S1 images is shown in Fig. A1. We did not apply
any radiometric terrain correction as the visible topography helps to identify the avalanche path direction.

For orthorectification we used the Swiss elevation model SwissAlti3D (2013) downsampled from 2 m to 30 m resolution.
We noticed, however, that despite of using the same DEM and output resolution, sharp topographic features seem to be better
orthorectified with the GAMMA software which might use a more precise spatial interpolation. The radar images were orthorectified to a resolution of 5x5 m (TSX) and 15x15 m (S1) and the backscatter signal in dB was saved to geotiff files. The
exact radiometric normalization is irrelevant, because we did not apply any radiometric terrain correction (Small, 2011) and
different ellipsoidal corrections ($\sigma_\mathrm{E}^0, \gamma_\mathrm{E}^0$) differ only by almost constant factors. Since the TSX data was acquired with a single
polarization (the co-polar channel HH) we also used only the co-polar channel (VV) of the two available polarizations of S1 to
obtain a fair comparison. For the multiorbital composites, we used both polarizations of S1 (VV and VH).

### 4.2 Single image avalanche detection

For avalanche detection by visual inspection in single images, areas with radar shadow and layover were masked out. The
images (as shown in Fig. 4a) were then systematically searched for bright features matching the tongue shaped geometry of the
avalanche deposition zone. Potential avalanches were manually contoured to create an avalanche mask. For uncertain cases, a
topographic relief map was used to decide if the identified shape corresponds to a possible flow path.

### 4.3 Two-image composite avalanche detection

Since detection in single images is difficult, we composed two consecutive acquisitions from the same orbit to an RGB change
detection image. To correct for large-scale backscatter changes due to wet snow a 500 m highpass filter was applied to the
backscatter difference between the two images. Examples for TSX and S1 are shown in Figs. 4b and 4c. To create the images,
the backscatter intensities in dB were normalized by clipping the lower and upper 1%. Consecutive images were then stored
in the channels [R, G, B] = [img2, img1, img1] so that backscatter changes are well visible by the red-cyan contrast: increased
backscatter appears red, decreased backscatter appears light blue (cyan), and unchanged backscatter appears gray.

The RGB change detection images allows for a temporal classification of avalanches into three classes: *new, old*, and *unsure*.
*New* avalanches appear red and are therefore assumed to have occurred between the first and the second acquisition. *Old*

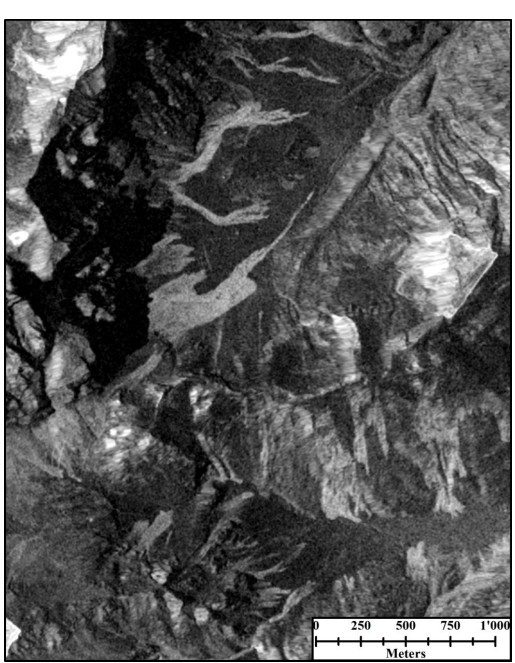

(a) Subset of single radar image TSX(01-11).

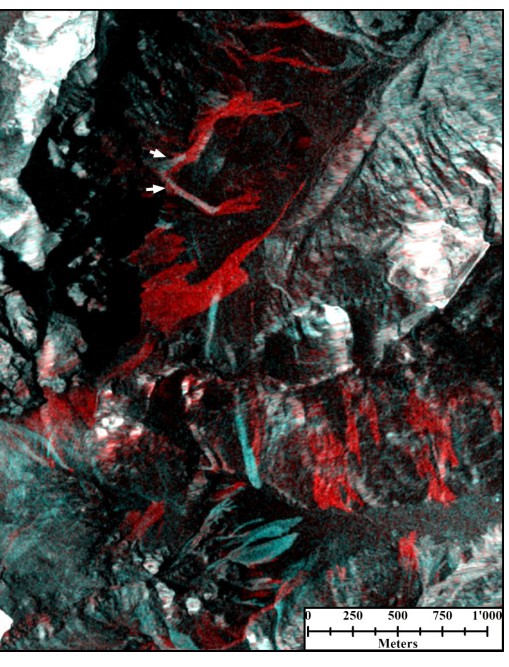

(b) Subset of change detection image tsx(12-31 / 01-11).

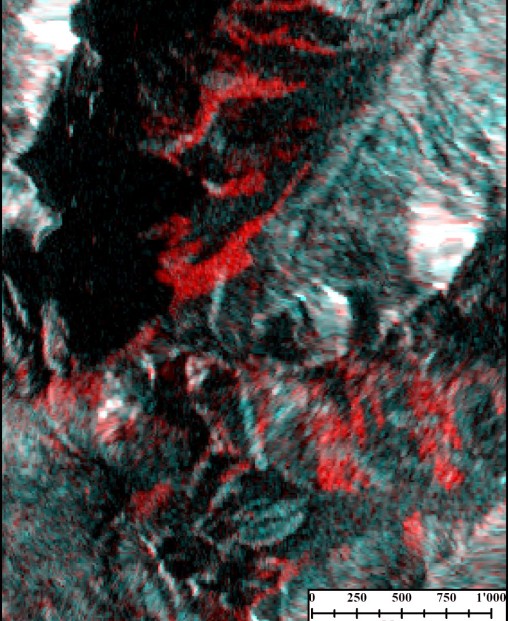

(c) Subset of change detection image S1(12-31 / 01-12).

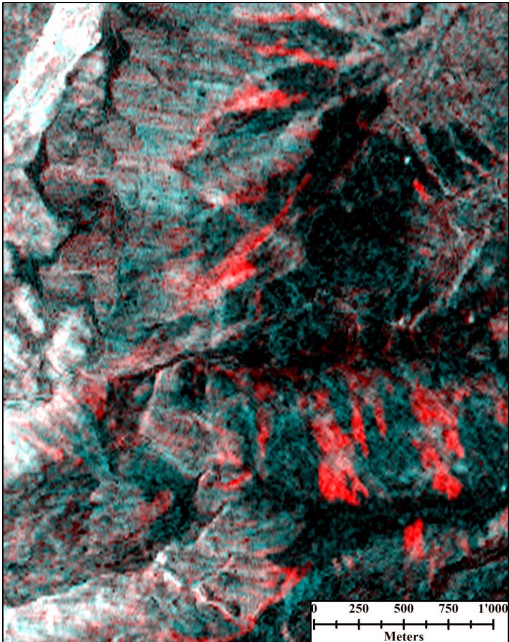

(d) Subset of S1 multiorbital change detection image from the two
data sets 2017-12-28 – 2018-01-01 vs. 2018-01-09 – 2018-01-12.

**Figure 4.** Subsets of the study area: (a) single radar image, (b,c) TSX and S1 change detection images, and (d) S1 multiorbital composite. The radar view direction is always from the left. Arrows in (b) indicate old avalanches overrun by new ones. All TerraSAR-X and Copernicus Sentinel data (2019) were orthorectified with the swissALTI3D © 2019 swisstopo (JD100042), reproduced with the authorisation of swisstopo (JA100120).
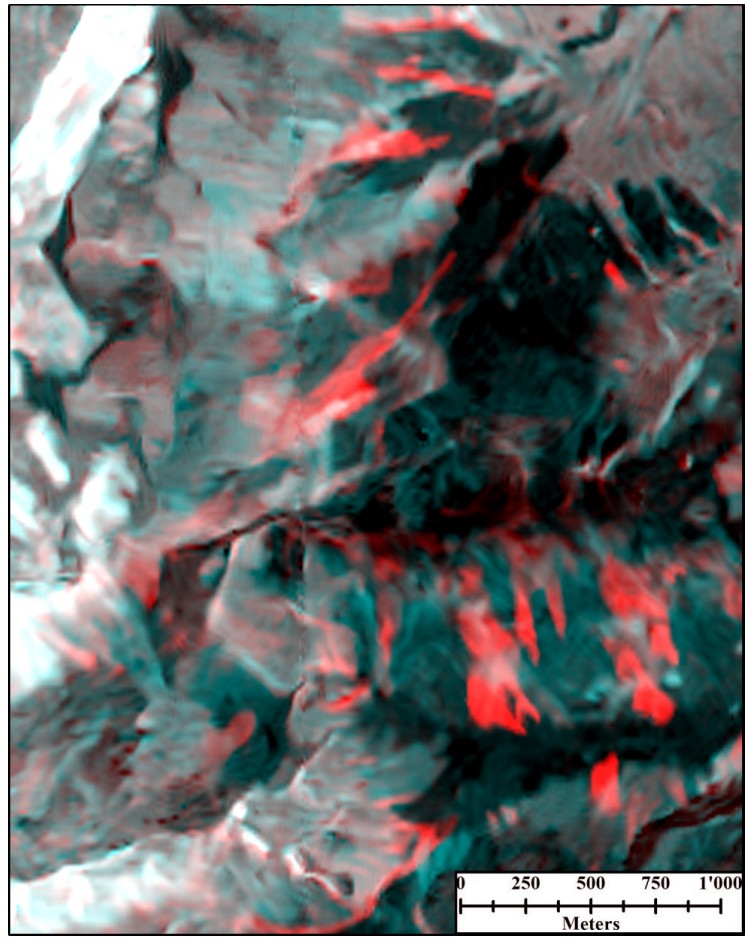

**Figure 5.** Multiorbital S1 change detection image as in Fig. 4d but with a non-local mean filter applied. Orthorectified with the swissALTI3D © 2019 swisstopo (JD100042), reproduced with the authorisation of swisstopo (JA100120). The corresponding full scene is shown in Fig. 9 and covers entire Switzerland.

avalanches appear blue are therefore assumed to have occurred before the first acquisition. Bright features with an unchanged

145  backscatter intensity appear almost white and are classified as *unsure* if they look like avalanches.

## 4.4 Multiorbital composite image for Switzerland

In contrast to the limited availability of TSX data, the free and systematic availability of S1 radar images and the short revisit period of only six days over central Europe allows for creation of an RGB composite change detection image covering entire Switzerland. Therefore, 12 images, acquired between 2017-12-28 and 2018-01-01 from different orbits, were combined into

150  an image before the first avalanche event (Jan 4th). Another 12 images, acquired between 2018-01-09 and 2018-01-12 with an identical imaging geometry, were used for the post-avalanche event image. The images (listed in Table A1) were preprocessed





according to Sect. 4.1. To reduce speckle we averaged both polarizations and weighted the cross-pol channel (VH) by the ratio $a$ of the co- and cross-pol backscatter intensities averaged over the entire scene:

$$S = \frac{S_{\mathrm{VV}} + a\,S_{\mathrm{VH}}}{1+a} \quad \text{with} \quad a = \frac{\langle S_{\mathrm{VV}} \rangle}{\langle S_{\mathrm{VH}} \rangle} \tag{2}$$

155      Then, the weighted mean was converted to dB and scenes from different ascending and descending orbits were averaged in overlapping areas. Thereby, a relatively homogeneous bright image is generated where layover areas lighten up the relatively dark slopes facing away from the radar without screening much of the contained details (Fig. 4d). To further reduce noise but to preserve edges in the mosaic images, we applied a non-local mean filter (Jin et al., 2011; Condat, 2010). The filtered image is shown in Fig. 5.

### 4.5    Relative brightness of snow avalanches

To analyze the brightness of avalanches relative to their surrounding, we calculated the ratio of the mean backscatter signal of an avalanche area and its surrounding area. Therefore, a visually determined avalanche mask was dilated once by 9 and once by 18 pixels. The difference of the two masks defines the surrounding. For the avalanche mask, the visual avalanche mask was eroded by 3 pixels to reduce manual contouring errors. To obtain statistically significant results we calculated the backscatter ratios only for avalanches and surrounding areas larger than 100 pixels.

### 4.6    Automated avalanche detection

As manual avalanche mapping is time consuming, a reliable automation of this process would make the mapping data quickly available for further application. Therefore, different attempts have been made to automatically detect avalanches mainly on the two satellite platforms S1 (Vickers et al., 2016; Wesselink et al., 2017; Abermann et al., 2019), and Radarsat-2 (Hamar et al., 2016; Wesselink et al., 2017). The general workflow in these papers is quite similar to ours. All methods are based on two-image change detection, application of various masks (layover, shadow, water bodies, forest), thresholding and filtering of extracted avalanche properties.

     In addition to a shadow and layover mask, we applied a slope dependent mask to limit the detection to potential avalanche deposition zones for which we expect the strongest backscatter change. By definition, friction is larger in the deposition zone than the downhill-slope force. Therefore, slopes with an inclination larger than 35°, which typically occur in the zone of origin, are masked out (Bühler et al., 2009).

     For noise reduction but to preserve avalanche edges, a $5 \times 5$ px median filter was applied to the backscatter difference images in dB. As avalanches should have a well defined edge, an edge mask was generated by applying a Sobel filter with a $5 \times 5$ kernel to the median filtered difference image.

     In the median filtered difference image, from all pixels brighter than a threshold of 4 dB, the brightest 5% were considered as the mask of potential avalanches. To remove isolated bright pixels from the mask, we determined around each continuous area an ellipse and removed areas with a major axis shorter than 15 pixels. Additionally, only potential avalanches for which more than 10 pixels intersect with the edge mask were considered for the final avalanche mask.





**Table 2.** Number of manually detected avalanches in TSX single images acquired after the first and after the second avalanche event.

| TSX image | total number of found avalanches |
| --- | --- |
| TSX(01-11) | 142 |
| TSX(02-02) | 120 |

## 5   Results

None of the mapping results obtained from TSX, S1, or SPOT-6 can be considered as real ground truth and different avalanches or avalanche shapes were detected with the different methods and satellites. Also, ambiguous relations can exist when a single large avalanche in one mapping result appears as multiple smaller avalanches in another mapping result. This makes the evaluation of binary classifies (e.g. probability of detection or false discovery rate) difficult or even impossible. We refrained from using a pixel-to-pixel comparison which would have demanded a manual mapping precision on the pixel level which

contradicts the subjective mapping by an experienced expert who sometimes estimates an avalanche outline from discontinuous avalanche patches.

As a remedy we compare results from two data sets A and B by reciprocal counting of avalanches which overlap in both data sets (considered as "found") and avalanches which do not overlap (considered as "not found"). These numbers differ depending on the direction in which the comparison is done (A→B or B→A). Depending which data sets is considered as ground truth,

avalanches which were "not found" can be either regarded as false negative alarms (missed) or as false positive alarms (false alarm).

For conciseness we refer in the following sections to the two single radar images from 2018-01-11 and 2018-02-02 as TSX(01-11) and TSX(02-02). Similar, we refer to the corresponding TSX RGB change detection images as tsx(12-31/01-11) and tsx(01-11/02-02) and to the S1 change detection image 2017-12-31 vs. 2018-01-12 as S1(12-31/01-12).

### 5.1   TSX single image

In the first image TSX(01-11), acquired 7 days after the first avalanche event, in total 142 avalanches were detected by visual inspection. Figure 4a shows a subset of the analyzed scene. Some avalanches can be clearly identified by their bright backscatter signal and their tongue-like shape.

In the second image, TSX(02-02), acquired 10 days after the second avalanche event, a total of 120 avalanches were detected.

Figure 6 shows a subset of the analyzed scene with an overlay of the manually generated avalanche mask. For this image, an unambiguous identification of avalanches was difficult. Because of the heavy avalanche activity on Jan 22/23[rd], multiple small new avalanches could have formed larger connected areas or even run over the same area multiple times such that the individual avalanches could not be identified. Because 10 days elapsed since the main avalanche event, avalanches progressively lost contrast to the surrounding snow due to loss of the surface roughness by surface melt, windblown snow or fresh snow.

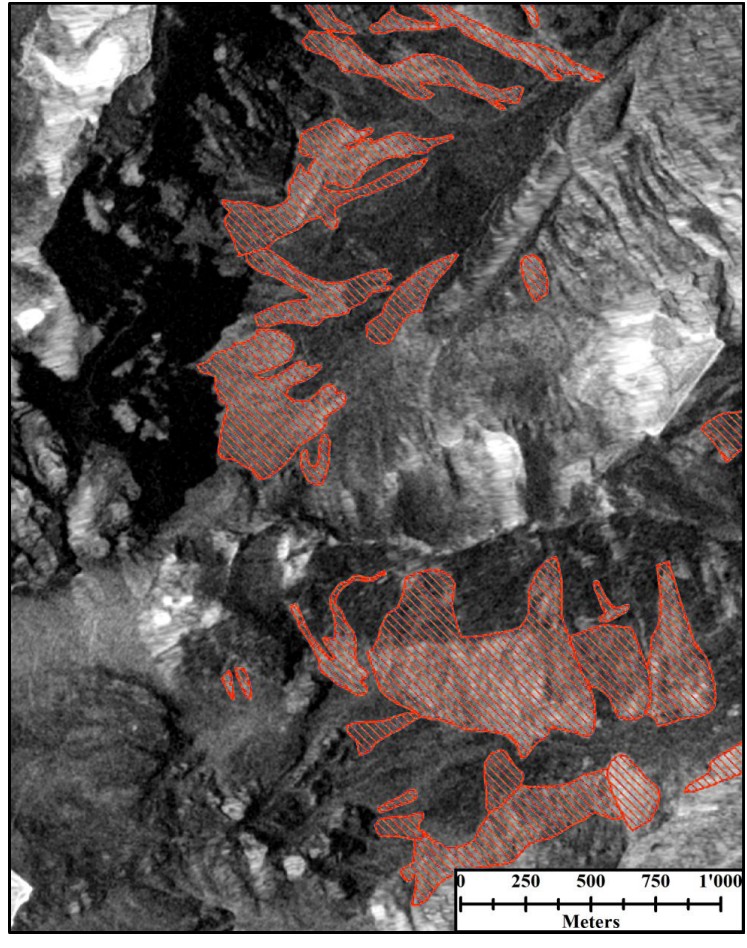

**Figure 6.** Single TSX backscatter image from 2018-02-02 (after the second avalanche event) including manually masked avalanches. Orthorectified with the swissALTI3D © 2019 swisstopo (JD100042), reproduced with the authorisation of swisstopo (JA100120).

## 5.2 TSX change detection

In the change detection image tsx(12-31/01-11), covering the first avalanche period, a total of 267 avalanches were manually detected. As detailed in Table 3, 164 avalanches were classified as *new* and 19 were classified as *old* avalanches. For 84 avalanches a clear assignment to *new* or *old* was not possible. Therefore, they were assigned to the class *unsure*. For example, in the upper part of Fig. 4b, two large new avalanches completely covered two small old avalanches (indicated by arrows). Therefore, the backscatter signal did not change and these old avalanches were classified as *unsure* (though they could be classified as *old* using spatial context information).

In the change detection image tsx(01-11/02-02), covering the second avalanche period, a total of 351 avalanches were detected, composed by 170 *new* avalanches, 35 *old* ones and 146 *unsure* cases. Most of these *unsure* avalanches were actually



**Table 3.** Number and classification of manually detected avalanches from TSX change detection images which cover the first and the second avalanche period.

| change detection image | total | *new* | *unsure* | *old* |
|:---:|:---:|:---:|:---:|:---:|
| tsx(12-31 / 01-11) | 267 | 164 | 84 | 19 |
| tsx(01-11 / 02-02) | 351 | 170 | 146 | 35 |

classified as *new* after the first avalanche period but overrun by new avalanches during the second avalanche period (compare
Fig. A2 with Fig. A3). Therefore, the number of *old* avalanches seems to remains low.

### 5.3 TSX change detection compared to single images

The TSX change detection images indicates a significantly enhanced sensitivity to avalanches compared to single TSX images. It seems that about twice as much avalanches (of class *new* and *unsure*) have been detected (248 vs. 142 and 316 vs. 120 avalanches, Table 2 vs. Table 3). However, around 90% of the avalanches detected in the single image overlap with one or
more avalanches classified as *new* or *unsure* in the change detection image. The better differentiation into multiple small avalanches is the main reason why the detection numbers in the change detection image are higher.

As detailed in Table 4a, when counting how many of 142 avalanches from TSX(01-11) overlap with avalanches from tsx(12-31/01-11), we found that 71% (101 avalanches) were also found as *new* and 12 avalanches were not found. Vice versa, 32% (53/164 avalanches) classified as *new* by change detection were not found in the single image (Table 4b).
A similar result is obtained for the 120 avalanches from the second single image TSX(02-02). As detailed in Table 4c, 85 of 120 avalanches (71%) were also detected as *new* in the change detection image and 6 avalanches were not found. Vice versa, 31% (52/170) of avalanches classified as *new* were not detected in the single image (Table 4d).

### 5.4 TSX compared to optical SPOT-6

The SPOT-6 images were acquired immediately after the second avalanche event in the morning of Jan 24[th]. Unfortunately due
to the 11 day revisit time of TSX, the first available TSX image from after the event was acquired 9 days later in the evening of Feb 2[nd]. During this 9 days surface melt occurred followed by about 20 cm of new snow on Feb 1[st]. Due to changing snow properties the contrast between avalanches and the surrounding snow has very likely decreased.

Nevertheless, as detailed in Table 5a, we found that 68% (85/120) of the avalanches detected in TSX(02-02) were also detected in the SPOT-6 image. Interestingly, of the remaining third (35/120) the majority (28 avalanches) were located in the
cast shadow.

Similar, for the change detection image tsx(01-11/02-02) Table 5b shows that 68% (215/316) of the avalanches detected as *new* or *unsure* were also detected in the SPOT-6 image. The remaining 101 were not detected, again, the majority of them (84) were located in the cast shadow.
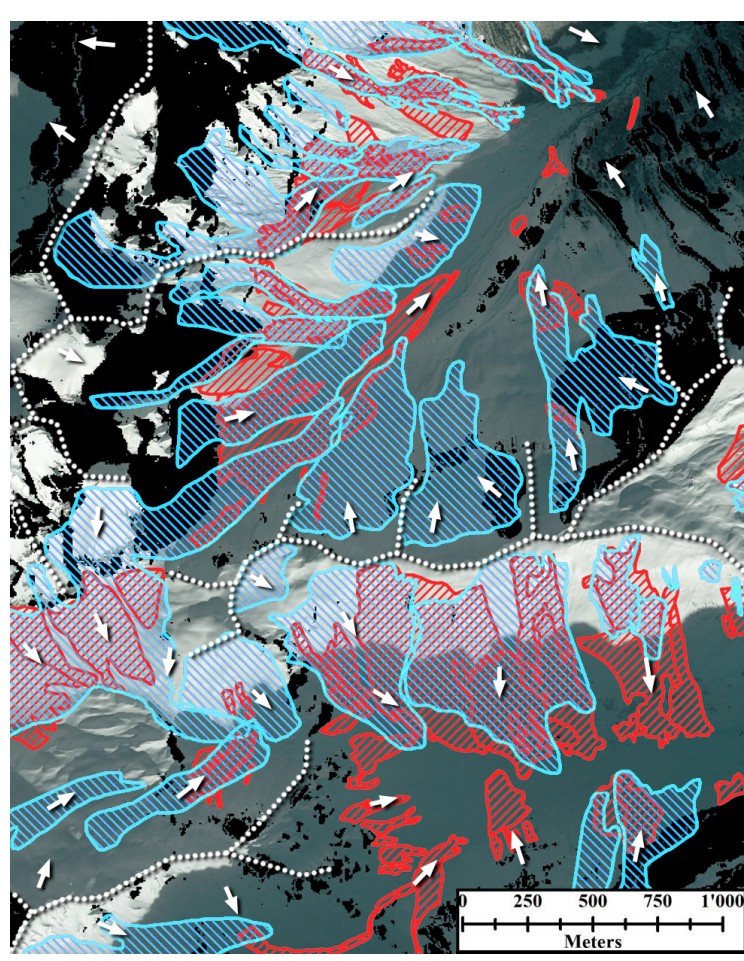

**Figure 7.** Manually mapped avalanches (blue) from the SPOT-6 image 2018-01-24 (background) vs. change detection results from tsx(01-11/02-02) (red, all classes). Orthorectified with the swissALTI3D © 2019 swisstopo (JD100042), reproduced with the authorisation of swiss-topo (JA100120). Radar shadow is added in black. Dots show mountain ridges and arrows the down-slope direction.





**Table 4.** Number of avalanches in single image which correspond to avalanches in TSX change detection image (a, c) and reverse correspondence of *new* avalanches from TSX change detection (b, d).

| (a) | of TSX(01-11) | | → | | tsx(12-31 / 01-11) | |
|---|---|---|---|---|---|---|
| | total | *new* | *unsure* | *old* | not found | |
| | 142 | 101 | 22 | 7 | 12 | |

| (b) | of *new* in tsx(12-31 / 01-11) | → | TSX(01-11) |
|---|---|---|---|
| | total | found | not found |
| | 164 | 111 | 53 |

| (c) | of TSX(02-02) | | → | | tsx(01-11 / 02-02) | |
|---|---|---|---|---|---|---|
| | total | *new* | *unsure* | *old* | not found | |
| | 120 | 85 | 27 | 2 | 6 | |

| (d) | of *new* in tsx(01-11 / 02-02) | → | TSX(02-02) |
|---|---|---|---|
| | total | found | not found |
| | 170 | 118 | 52 |

Vice versa, 44% (125/286) of the optically detected avalanches were also found in the TSX change detection image (Table 5c) but more than half of the optically detected avalanches were not found. 20% (57/286) could not be found because they are located in the radar shadow and 36% (104/286) had a too low contrast to be visible with radar.

With the temporal information from radar change detection the 125 avalanches detected with SPOT-6 but also with TSX (Table 5c) can be further classified into 27 *new*, 38 *unsure*, and 6 *old* avalanches. The remaining 54 avalanches could not be uniquely classified, because they cover areas which were differentiated into multiple different classes by radar whereas such a temporal classification is difficult with single SPOT6 images (Bühler et al., 2019).

Figure 7 shows a subset of the SPOT-6 images and visualizes the manually mapped avalanches. Especially in the lower part of the image, in the cast shadow, many small radar-detected avalanches (red) were not found in the optical analysis (blue). With radar, avalanches could generally not be detected in the radar shadow or layover (added with black) but also many other avalanche were missed by radar.

## 5.5 TSX compared to S1 change detection

To asses the added value of high resolution TSX images compared to medium resolution S1 images we compared the corresponding mapping results. Images from the first avalanche period were chosen to simplify counting because of less overlapping old and new avalanches. In the S1 change detection image S1(12-31 / 01-12) a total of 89 *new*, 13 *unsure*, and 16 *old* avalanches were found. Compared to TSX, the S1 change detection image shows a significantly lower resolution (Fig. 4b vs. Fig. 4c) such that smaller avalanches are more likely not to be mapped (Fig. 8).

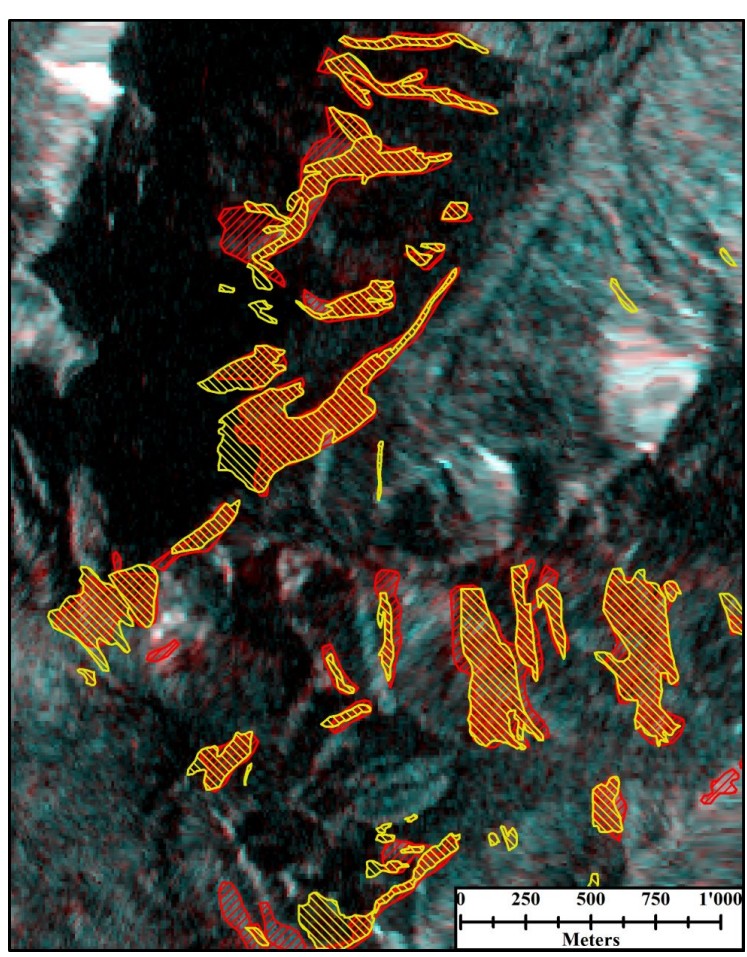

**Figure 8.** Manually mapped *new* avalanches (in red) from the change detection image S1(12-31/01-12) (background) compared to manually mapped *new* avalanches from tsx(12-31/01-11) (yellow). No mask is shown for avalanches classified as *old* or *unsure*. Orthorectified with the swissALTI3D © 2019 swisstopo (JD100042), reproduced with the authorisation of swisstopo (JA100120).





**Table 5.** Number of avalanches detected in single (a) and change detection (b) TSX radar images compared to avalanches which were also detected in the optical SPOT-6 data. (c): reverse correspondence of avalanches from SPOT-6 to *new* and *unsure* avalanches from the radar change detection image. Avalanches which were not found are grouped depending on if they are located in the cast shadow (a,b) or in the radar shadow (c).

| (a) | of TSX(02-02) | → | SPOT-6 (01-24) | |
|---|---|---|---|---|
| | total | found | not found | (in / not in cast shadow) |
| | 120 | 85 | 35 | (28 / 7) |
| (b) | of *new/unsure* in tsx(01-11 / 02-02) | | → SPOT-6 (01-24) | |
| | total | found | not found | (in / not in cast shadow) |
| | 316 | 215 | 101 | (84 / 17) |
| (c) | of SPOT-6 (01-24) | → | *new/unsure* in tsx(01-11 / 02-02) | |
| | total | found | not found | (in / not in radar shadow) |
| | 286 | 125 | 161 | (57 / 104) |

As detailed in Table 6, from the 89 *new* avalanches, 83 were also found by TSX. They correspond to 76 *new* and 7 *unsure* avalanches; 6 avalanches were not found. Vice versa, two thirds (104/164) of the avalanches found in tsx(12-31/01-11) correspond to the 83 avalanches also found with S1. One third (60/164) was not found, mostly because they were too small to be detected with S1.

**Table 6.** (a) Number of manually detected *new* avalanches in S1(12-31/01-12) which were also detected as *new* or *unsure* in the change detection image tsx(12-31/01-11). (b) reverse correspondence.

| (a) | of *new* in S1(12-31 / 01-12) | → | tsx(12-31 / 01-11) | |
|---|---|---|---|---|
| | total | found | (*new / unsure*) | not found |
| | 89 | 83 | (76 / 7) | 6 |
| (b) | of *new* in tsx(12-31 / 01-11) | → | S1(12-31 / 01-12) | |
| | total | found | (*new / unsure*) | not found |
| | 164 | 104 | (100 / 4) | 60 |

### 5.6 Multiorbital S1 change detection composite

By combining S1 acquisitions from multiple ascending and descending orbits, we minimized areas affected by radar shadow and layover. The multiorbital change detection composite is shown in Fig. 9 and covers entire Switzerland during the first avalanche period. In the full, non-local mean filtered 15 m-resolution image, which is available online (Leinss et al., 2019), we counted 7361 avalanches. We found that avalanches reaching below the wet snow line were much better visible than avalanches

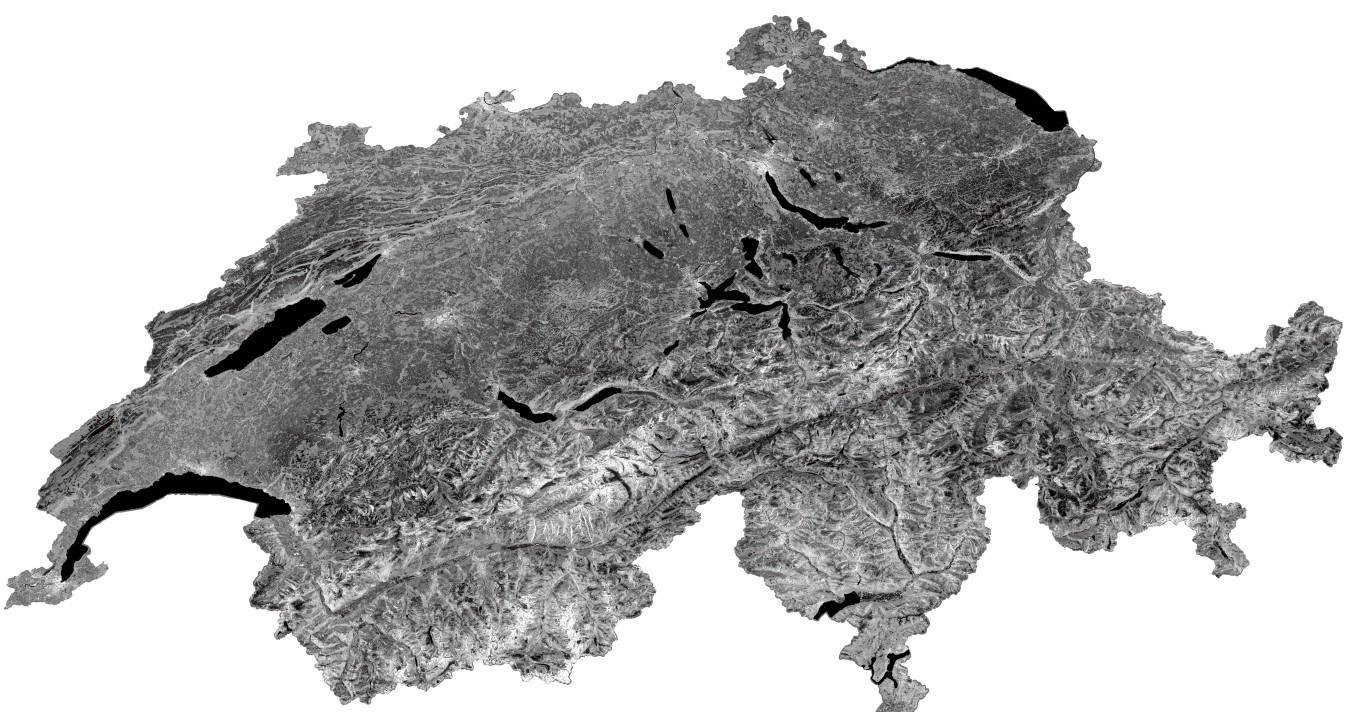

**Figure 9.** In the 15 m-resolution multiorbital S1 change detection mosaic, covering entire Switzerland for the first avalanche period around Jan 4[th], we counted 7361 *new* avalanches. When zooming into the image, many avalanches are visible in red. The image is combined from each 12 acquisitions from 2017-12-28 until 2018-01-01 and from 2018-01-09 until 2018-01-12 and is available online (Leinss et al., 2019). All Copernicus Sentinel scenes (2019) were orthorectified with the swissALTI3D © 2019 swisstopo (JD100042), reproduced with the authorisation of swisstopo (JA100120).

from the dry snow zone. Figure 4d shows a subset which illustrates the mitigation of shadow and layover, the speckle reduction and the enhanced resolution compared to the single orbit S1 image in Fig. 4c.

The comparison of the multiorbital S1 mapping results with the high resolution TSX data is detailed in Table 7. In the study area a total of 136 *new* avalanches were manually detected in the multiorbital image (S1-MO). Of these, 104 avalanches match with avalanches detected in the corresponding single orbit TSX change detection scene (95 of them with *new* avalanches, 9 with *unsure*), whereas 32 avalanches were not found with TSX. 17 of the 32 avalanches could not be detected because they are in the shadow/layover areas of TSX. Vice versa, 110 of 164 TSX avalanches were also detected in the multiorbital S1 composite whereas 54 TSX avalanches were not detected.

**5.7 Automated avalanche detection**

For the implemented automatic avalanche detection algorithm we chose a threshold of 4 dB for the relative brightness of avalanches which corresponds to the upper 82% of the avalanche brightness distribution shown in Fig. 10a. The figure is based





**Table 7.** (a) Number of the *new* avalanches in the S1 multiorbital change detection image (S1-MO) compared to avalanches in the TSX change detection image. Reverse correspondence in (b).

| (a) | *new* in S1-MO(12-28+4d / 01-09+4d) →tsx(12-31 / 01-11) | |
| --- | --- | --- |
| total | found (*new/unsure*) | not found (in/not in shadow) |
| 136 | 104  (95 / 9) | 32   (17 / 15) |

| (b) | *new* in tsx(12-31 / 01-11) →S1-MO(12-28+4d / 01-09+4d) | |
| --- | --- | --- |
| total | found | not found |
| 164 | 110 | 54 |

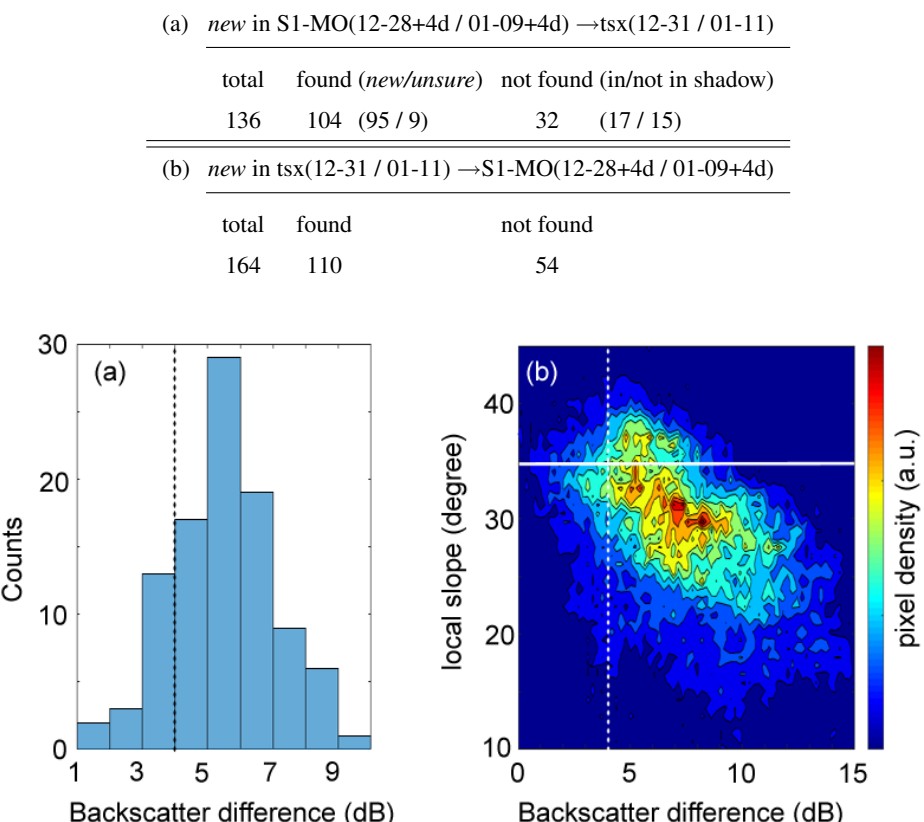

**Figure 10.** (a) Histogram of the mean relative brightness of avalanches for manually mapped *new* avalanches in tsx(12-31/01-11). (b): Relative brightness of the avalanche pixels in relation to the local slope angle. Lines indicate the thresholds for the backscatter difference and the slope-dependent mask.

on 99 of 164 *new* avalanches which cover more than 100 pixels (Sect. 4.5) and which were selected from tsx(12-31/01-11). The threshold to mask out areas steeper than $35°$ (Sect. 4.6) is supported by the slope-dependent distribution of avalanche pixels in Fig. 10b. With these settings, the automatic methods identified about two thirds of the manually identified avalanches in the same image pair. Here we considered the manually determined avalanche mask as a proxy for the true extend of the deposition zone. We are aware that the significance of such a comparison is limited. Nevertheless, the advantage of this comparison is that the performance of the detection algorithm is directly compared to the results of a human avalanche mapping expert.

For the first image pair tsx(12-31/01-11) Table 8a details that 110 of 164 manually mapped *new* avalanches were also found with the automated detection whereas 54 were not found. As shown in Fig. 11, these "missed" avalanches are often very small
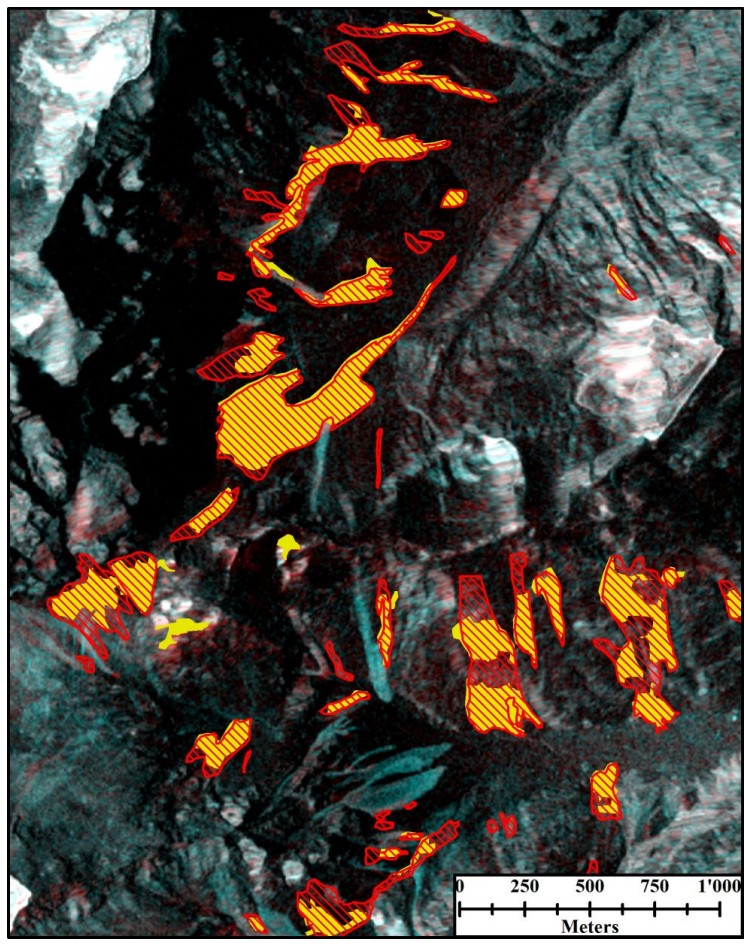

**Figure 11.** Comparison between detected *new* avalanches when manually mapped (red) and automatically detected (yellow) in the TSX acquisition pair 2017-12-31 vs. 2018-01-11. Orthorectified with the swissALTI3D © 2019 swisstopo (JD100042), reproduced with the authorisation of swisstopo (JA100120).

avalanches which were filtered out by the algorithm. Vice versa, of 138 automatically detected avalanches 21 were not found manually (Table 8b).

When considering the total number of manually detected avalanches (164) as truth one can assign avalanches which were also found automatically to *true positive* (TP = 110), i.e. correctly detected. The remaining avalanches, which were not automatically detected, are then assigned to *false negative* (FN = 54), i.e. incorrectly rejected. With this assumption the probability of detection (POD) and the miss rate or false negative rate (FNR) can be calculated:

$$\text{POD} = \frac{\text{TP}}{\text{TP} + \text{FN}} \qquad \text{and} \qquad \text{FNR} = \frac{\text{FN}}{\text{TP} + \text{FN}} = 1 - \text{POD} \qquad (3)$$

Further, one can assign automatically detected avalanches which were not manually found to *false positive*s (FP = 21), i.e. incorrectly detected. When assuming that the number of correctly detected avalanches is given by TP = 110, the false discovery





rate (FDR) reads

$$FDR = \frac{FP}{FP + TP} \qquad (4)$$

With that one obtains a POD = 67 %, a miss rate FNR = 33 % and a false discovery rate FDR = 16 % for tsx(12-31/01-11).

For the second image pair tsx(01-11/02-02) only 82 of 170 manually detected *new* avalanches were automatically found whereas 88 were not found (Table 8c). Vice versa, 54/179 automatically detected avalanches were not found manually (Table 8d). Assuming again that the manually detected avalanches are the true avalanches one obtains a POD of 48 %, a FNR of 52 %, and a FDR = 40 %. The results are expected to be worse compared to the first period, because mapping of *new* avalanches 305   was very difficult for the second period where many old and new avalanches overlapped such that many *unsure* cases occurred for which the backscatter signal changed less than the threshold of 4 dB.

The automated algorithm was also run on the S1 images pair S1(12-31/01-12). As detailed in Table 8e, 68 of 89 manually detected *new* avalanches were also found automatically whereas 21 were not found. Vice versa, of 92 automatically mapped avalanches 72 were also found manually and 20 were not found (Table 8f), resulting in a POD = 76%, a FNR = 24%, and a 310   FDR = 23%.

The higher POD and lower FNR for S1 compared to TSX indicates only, that with S1 the automatic method can detect a larger fraction of the manually detected avalanches. It does not indicate that results obtained from S1 are better compared to TSX data where in total more avalanches were detected.

## 6   Discussion

**6.1   Advantage of radar change detection images**

In both high resolution TSX stripmap radar images more than one hundred avalanches have been manually detected by visual inspection of single radar images within the analyzed area covering 132 km². However, a clear advantage has been found when using successive images for change detection where about twice as much avalanches could be differentiated (Sect. 5.3).

Additionally, change detection images provide a temporal information which makes it possible to differentiate relatively 320   clearly between new and old avalanches. Therefore, they can be much easier differentiated from bright regions of similar shape but which are in reality erosion features like scree or talus deposits and which do not significantly change their backscatter behavior in time (therefore classified as *unsure*). Still, the differentiation of strongly overlapping avalanches is difficult and requires a high spatial and temporal resolution. In the extreme case, a temporal resolution of seconds or minutes would be required to temporally resolve individual avalanches. The fast repeat times of current radar satellites, like 6 days when combining 325   the two S1 satellites is a major advantage compared to other mapping methods. For example, compared to the 11 days of TSX, the temporal resolution of S1 is almost twice as high. Due to clouds, optical data is not reliably available and temporally dense time series cannot exist for weather conditions with heavy snow fall and strong winds during which the highest avalanche risk occurs.




**Table 8.** Number of automatically detected *new* avalanches compared to the number of manually detected *new* avalanches from the same image pair.

| (a) | man:tsx(12-31 / 01-11) | $\rightarrow$ | auto:tsx(12-31 / 01-11) | | |
|---|---|---|---|---|---|

| total | found | POD | not found | FNR |
|---|---|---|---|---|
| 164 | 110 | 67% | 54 | 33% |

| (b) | auto:tsx(12-31 / 01-11) | $\rightarrow$ | man:tsx(12-31 / 01-11) | |
|---|---|---|---|---|

| total | found | not found | FDR |
|---|---|---|---|
| 138 | 117 | 21 | 16% |

| (c) | man:tsx(01-11 / 02-02) | $\rightarrow$ | auto:tsx(01-11 / 02-02) | | |
|---|---|---|---|---|---|

| total | found | POD | not found | FNR |
|---|---|---|---|---|
| 170 | 82 | 48% | 88 | 52% |

| (d) | auto:tsx(01-11 / 02-02) | $\rightarrow$ | man:tsx(01-11 / 02-02) | |
|---|---|---|---|---|

| total | found | not found | FDR |
|---|---|---|---|
| 179 | 125 | 54 | 40% |

| (e) | man:S1(12-31 / 01-12) | $\rightarrow$ | auto:S1(12-31 / 01-12) | | |
|---|---|---|---|---|---|

| total | found | POD | not found | FNR |
|---|---|---|---|---|
| 89 | 68 | 76% | 21 | 24% |

| (f) | auto:S1(12-31 / 01-12) | $\rightarrow$ | man:S1(12-31 / 01-12) | |
|---|---|---|---|---|

| total | found | not found | FDR |
|---|---|---|---|
| 92 | 72 | 20 | 23% |

## 6.2 Optical mapping vs. radar change detection

Regardless of the advantages of radar change detection, the spatial resolution of optical sensors is better compared to radar sensors of the same nominal resolution because the intrinsically coherent SAR imaging method makes radar speckle unavoidable and requires spatial or temporal averaging. Furthermore, the resolution of TSX and S1 is not good enough to recognize flow structures of the avalanche surface which are well visible in the optical SPOT-6 images (Bühler et al., 2019).

With TSX change detection images we have mapped a similar number of avalanches (316) compared to the results from 335 optical SPOT-6 images (286 avalanches) within the analyzed area. However, the mapped avalanche outlines differ relatively strongly and only 68% of the radar detected avalanches overlap with avalanches found with the optical data or inversely, only 44% of optically detected avalanches were also found by radar. The differences of the avalanche outlines could also be partially attributed to the estimation of avalanche outlines by the person mapping the avalanches.





The fact that a larger fraction (68% vs. 44%) of radar-detected avalanches matches with optically detected ones results from
the better differentiation of adjacent avalanches into multiple classes (*new, old, unsure*) which have been often mapped as
one large avalanche with optical data. When multitemporal optical data is available, a temporal differentiation is also possible
(Bühler et al., 2019) which, however, was done for a different region than our analyzed area.

From the analysis of avalanches detected by radar but not found in the optical SPOT-6 image, we found that more than 80%
of these avalanches were located in the cast shadow. It seems that these avalanches are not easy to detect in the optical images
whereas they are well visible in radar images. Similar, in radar images no information is available from the radar shadow and
very poor information is available from layover areas, however, only 35% of avalanches not found in the radar images (but
in optical) are located in the radar shadow or layover. We think it is an important result that not only radar acquisitions are
affected by (radar) shadow but that avalanche mapping results with optical data seem also to be deteriorated by the cast shadow
from high mountains.

Unfortunately, the direct comparison of SPOT-6 and TSX data is limited by several uncertainties: the acquisition date of the
SPOT-6 images (2018-01-24) was just between the two TSX images (2018-01-11 and 2018-02-02) and leaves 9 days where
additional avalanches could have occurred, considering about 20 cm of fresh snow on Feb $1^{st}$. Nevertheless, Fig. 2 indicates
that the biggest part of avalanches occurred before the SPOT-6 acquisition and only about 5% of avalanches occurred between
the SPOT-6 and the TSX acquisition from Feb $2^{nd}$. To confirm this we analyzed a multiorbital S1 change detection image
S1(01-24+01-28/01-30+02-03) and did not find any new avalanches in the study area. As the avalanche risk was slightly higher
(level 2–3) for north exposed slopes after Jan $24^{th}$ (SLF, 2018b, c) we cannot completely exclude that during this time an
unknown amount of avalanches too small to be detected by S1 could have still released in the case shadow.

### 6.3 TSX compared to S1 change detection

The comparison of TSX and S1 change detection images, both of them acquired for the first avalanche period with almost
identical orbits and acquisition times shows that the S1 satellites are a valuable source of radar images for avalanche mapping.
Though with S1 very small avalanches are likely to be missed. Still about two thirds of avalanches detected and classified as
*new* with TSX could also be detected with S1 (Sect. 5.5). Notably, 93% (83/89) of avalanches detected by S1 could also be
detected by TSX which reflects the agreement between TSX and S1 mapping results. Also, the shape of the avalanches masked
in S1 data is very similar to the one from TSX (Fig. 8). Therefore, we consider the reduced resolution and separability of
avalanches in S1 images to be much less relevant than the superior availability of S1 data.

### 6.4 Multiorbital composite

The combination of radar acquisitions from ascending and descending orbits minimizes areas affected by radar shadow and
layover and reduces speckle noise. Noise was further reduces by averaging both polarizations (VV, VH) available from the dual-
polarization S1 data. By combining the two orbit directions and the two (pairwise incoherent) polarizations, areas visible from
both orbits were imaged by 4 independent observations. This number can even increase to 6 or 8 observations when acquisitions
with different incidence angles (from the same orbit direction) overlap. Due to the 4–8 independent observations, spatial



multilooking (used for speckle reduction) could be reduced to $4\times1$ pixels to obtain a radiometric accuracy otherwise only possible with multilooking windows of $8...16\times2$ pixels. With this multiorbital averaging method, we estimate that an effective spatial resolution of about $20\times20\,\mathrm{m}$ was achieved (TSX: about $10\times10\,\mathrm{m}$ after multilooking). This resolution enhancement can be clearly observed when comparing Fig. 4c with Fig. 4d. Still, these values hold only for flat terrain. For geometric reasons the local resolution $\delta_{\mathrm{rg}} = \delta_{\mathrm{sr}}/\cos\theta$ of mountain slopes depends on the local incidence angle $\theta$ such that, compared to flat terrain, slopes facing off the radar show an increased resolution whereas slopes facing away from the radar show a significantly deteriorated image quality or even complete loss of resolution in the case of layover.

To combine multiple orbits we simply averaged the change detection radar images in dB which preserved the relative brightness and resolution of local features (avalanches). By the simple average slopes facing off the radar were mainly lightened up by the average with a bright layover area. We think that more advanced methods to merge radar images from multiple orbits, for example local resolution weighting (LRW) by Small (2012), should further improve avalanche mapping results.

From the comparison to optical data we also found that avalanches can be clearer identified in slopes facing off the radar compared to slopes which are facing towards the radar (but not yet in layover). As detailed in Sect. 3, we think that because of the more isotropic scattering of the rough surface of avalanches steep radar incidence angles should be used to enhance the local contrast to the surrounding snow. Therefore, slopes facing away from the sensor should be given more weight which is implicitly already done by LRW.

The additionally applied non-local mean filter further increased the visibility of avalanches (compare Fig. 4d with Fig. 5). The enhanced geometric and radiometric resolution through multiple orbits and the non-local-mean filter makes the separability of avalanches in S1 data almost equivalent to single-orbit and single-polarization TSX data (Sect. 5.6). This conclusion is also supported by the avalanche differentiation ratio of 1.06 (Table 9) which is discussed in Sect. 6.6.

### 6.5 Automated avalanche detection

For both, TSX and S1 images the implemented avalanche detection algorithm performs with reasonable results, at least when the number of overlapping avalanches is low. As shown in Fig. 11, the area of automatically detected avalanches (yellow) shows a good agreement with manually detected avalanches (red shading). However, the upslope parts of avalanches are often only fractionally detected because of their relatively low brightness. For a weakly visible starting or transition zone a human observer can conclude that it must belong to the below situated avalanche deposit. Also, by choosing a threshold of $4\,\mathrm{dB}$ already $18\,\%$ of the manually detected avalanches are likely to be missed (Fig. 10a). Further, minor parts of manually detected avalanches are located in slopes steeper than $35°$ (Fig. 10b) which were masked out by the automatic method.

### 6.6 Avalanche differentiation with different methods

The fact that no real ground truth exists makes a direct comparison of the different methods difficult. However, some methods show a much higher potential to differentiate avalanches than others. Because of the better differentiation, the quantitative comparison of found avalanches should be interpreted with the consideration that multiple avalanches can correspond to a single, big avalanche.



**Table 9.** Avalanche differentiation ratios between different satellite acquisitions and methods, and mutual miss-/false discovery rates.

| Set A | Set B | $\frac{N_{A\to B}}{N_{B\to A}}$ | $\frac{N_{A\neg B}}{N_A}$ | $\frac{N_{B\neg A}}{N_B}$ |
|---|---|---|---|---|
| tsx(12-31 / 01-11) | TSX(01-11) | 1.10 | 33% | 8% |
| tsx(01-11 / 02-02) | TSX(02-02) | 1.39 | 31% | 5% |
| tsx(12-31 / 01-11) | SPOT-6 (01-24) | 1.72 | 32% | 56% |
| tsx(12-31 / 01-11) | S1(12-31 / 01-12) | 1.25 | 37% | 7% |
| tsx(12-31 / 01-11) | S1 MO-1 | 1.06 | 33% | 24% |
| tsx(12-31 / 01-11) | manual vs. auto | 0.94 | 33% | 15% |
| S1(12-31 / 01-12) | manual vs. auto | 0.94 | 24% | 22% |
| tsx(01-11 / 02-02) | manual vs. auto | 0.66 | 52% | 30% |

As a proxy for the enhanced differentiation we define the ratio $N_{A\to B}/N_{B\to A}$ where $N_{A\to B}$ is the number of avalanches from data set A which were also found in the data set B, and inversely, $N_{B\to A}$ is the number of avalanches in B which were also found in A. Additionally, we define the ratio $N_{A\neg B}/N_A$ of avalanches found in A but not found in B relative to all avalanches found in A and analogue $N_{B\neg A}/N_B$. The meaning of the last two ratios depends on interpretation and correspond to the false discovery rate (FDR) under the assumption that B is considered as truth or alternatively to the false negative rate (FNR) if A is

considered as truth.

Table 9 lists the three ratios for different data sets. We interpret these numbers such that a differentiation ratio $\frac{N_{A\to B}}{N_{B\to A}} > 1$ indicates that set A provides spatially more detailed results than set B. An asymmetry between the last two columns indicates that one method detects more avalanches than the other method.

From the first two rows of Table 9, derived from *new* avalanches in Table 4, we conclude that with change detection

avalanches are better to differentiate and that more than 30% of *new* avalanches can be detected compared to mapping using single images. We also conclude that the miss-detection rate of single images is relatively low (5-8%).

From the comparison to SPOT-6, derived from Table 5, we infer that TSX change detection allows for a better differentiation of avalanches than single optical images. However, both methods show miss rates (and possibly some false detection) of 32 and 56% for avalanches which are not visible by the other method which indicates a certain complementarity of optical and

radar images for avalanche detection.

Compared to S1, the higher resolution of TSX allows for a 25% better differentiation and 37% more avalanches were detected (derived from Table 6). Still, the false discovery rate of S1 compared to TSX is quite low (7%).

Interestingly, the avalanche separability of the multiorbital S1 composite is very comparable to TSX single orbit change detection (1.06) while 33% or 24% of avalanches detected by one method are not visible with the other (derived from Table 7).

This, because TSX detects smaller avalanches, while the multiorbital methods detects also avalanches which are otherwise in the radar shadow or in slopes facing the radar.

Finally, the automatic methods detects larger avalanches fairly comparably to the manual method (derived from Table 8), however, weakly visible avalanches and small avalanches which have not been automatically detected cause a miss rate of about





30 %. The apparently lower differentiation of avalanche by manual analysis results from the fact that the automatic method
often detects multiple patches instead of a single avalanche (Fig. 11).

## 7    Conclusions

We studied the capabilities of the radar satellites TerraSAR-X (TSX) and Sentinel-1 (S1) to detect avalanches using single
radar images, two-image change detection images and multiorbital change detection composites. Manual avalanche mapping
results from the high- and medium resolution radar data (TSX, S1) and high resolution optical data (SPOT-6) were compared
to each other. An automatic detection method was developed and compared to the manual mapping results.

We conclude that both, TSX and S1 radar images can provide valuable, weather-independent information about avalanche
activity, even in difficult alpine terrain. We showed that avalanches can be manually identified in single radar images, but
the mapping precision is significantly enhanced and a temporal information is added when two consecutive radar images are
combined into an RGB change detection image. With that we found a fair agreement between TSX results and mapping from
single SPOT-6 images. Interestingly, many avalanches located in the cast shadow were not detected in the optical SPOT-6
image whereas they were clearly visible in a TSX image acquired 10 days later. With the automated detection algorithm we
found about 60–80% of the avalanches manually mapped in the same image, at least when no large number of old avalanches
were present.

Despite of the high resolution of TSX which allows for a more detailed avalanche mapping, the non-systematic acquisition
program and possibly high cost can be considered as drawback of TSX data. Also, with the maximal swath width of 30 km
in stripmap mode and a nominal revisiting period of 11 days, an operational use for avalanche mapping over Switzerland
with TSX is not feasible. However, high resolution radar from TSX can provide valuable information for validation of lower
resolution mapping results.

In contrast to TSX, and despite of its lower resolution, we found that S1 provides a convincing solution for systematic
avalanche mapping because of the total swath width of 250 km and the revisit period of 6 days when images from the same
orbit of both satellites (S1-A and S1-B) are combined. The results from Norway by Eckerstorfer et al. (2018) confirm this
conclusion. For a selected test site we detected with S1 about two thirds of the avalanches found with TSX but small avalanches
were often missed. With the combination of systematically available S1 acquisitions from different orbits and with different
polarizations we minimized not only areas located in radar shadow and layover but also enhanced the radiometric accuracy and
obtained a high spatial resolution of about $20 \times 20$ m. In the resulting change detection image covering entire Switzerland we
found in total 7361 new avalanches which occurred during an extreme avalanche period around January 4[th] 2018. However,
we suppose that mainly avalanches reaching below the wet snow line were detected and that likely many dry snow avalanches
were missed because of their lower contrast to the surrounding snow. We think that avalanche mapping can be even improved
with more advanced methods to combine different orbits, for example with local resolution weighing (Small, 2012). With that
slopes facing off the radar are weighted stronger which does not only enhance the resolution but should also increase the





avalanche visibility as the more omnidirectional scattering of the rough avalanche surface dominates the scattering of smooth snow slopes facing off the radar.

Although we could show that radar change detection mapping with TSX provides results comparable to optical SPOT-6 direct mapping, we note that our study focuses on the exceptionally warm January 2018 with frequent surface melt but also

with very intense snowfall periods. As the relative brightness of avalanches with respect to the surrounding snow depends on the water content and the amount of deposited snow, avalanches might be less visible during cold weather with little snowfall. Therefore, we think that the analysis of longer time series of radar based avalanche mapping is required to assess the detection rate of radar based methods under different weather conditions and in different regions.

*Data availability.* TerraSAR-X data are available from the archive https://terrasar-x-archive.terrasar.com. Copernicus Sentinel-1 data pro-
cessed by ESA have been downloaded from the Copernicus Open Access Hub: https://scihub.copernicus.eu and from the Alaska SAR Facility ASF DAAC 2018 https://www.asf.alaska.edu. The manual mapping results from the optical data and the Sentinel-1 change detection composite of Switzerland is available online (Hafner and Bühler, 2019; Leinss et al., 2019).

**Appendix A**

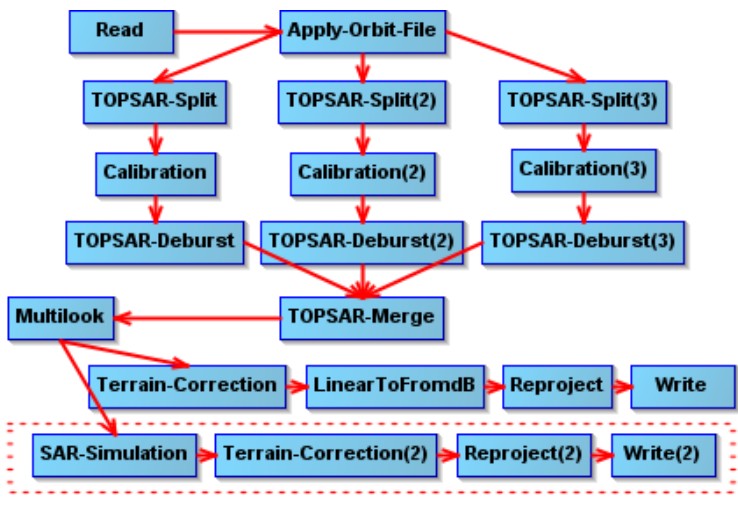

**Figure A1.** SNAP workflow to process S1 data. The red dashed box is used for creation of the layover and shadow map.

*Author contributions.* SL and RW wrote the manuscript, RW did the avalanches mapping and designed the automatic detection. SL co-
ordinated the study and implemented the nonlocal mean filter. SB, SH, SL preprocessed the radar images. YB initiated the study and complemented the manuscript.



**Table A1.** List of S1 acquisitions used for the multiorbital RGB change detection composite shown in Fig. 9.

| Satellite | Date | Time (UTC) | rel. orbit, direction |
|---|---|---|---|
| Sentinel-1B | 2017-12-28 | 05:42:17 | 139, descending |
| Sentinel-1B | 2017-12-28 | 05:42:43 | 139, descending |
| Sentinel-1A | 2017-12-29 | 05:34:45 | 66, descending |
| Sentinel-1A | 2017-12-29 | 05:35:10 | 66, descending |
| Sentinel-1B | 2017-12-30 | 05:26:02 | 168, descending |
| Sentinel-1B | 2017-12-30 | 05:26:27 | 168, descending |
| Sentinel-1A | 2017-12-30 | 17:23:14 | 88, ascending |
| Sentinel-1A | 2017-12-30 | 17:23:39 | 88, ascending |
| Sentinel-1B | 2017-12-31 | 17:14:13 | 15, ascending |
| Sentinel-1B | 2017-12-31 | 17:14:38 | 15, ascending |
| Sentinel-1A | 2018-01-01 | 17:06:47 | 117, ascending |
| Sentinel-1A | 2018-01-01 | 17:07:12 | 117, ascending |
| Sentinel-1B | 2018-01-09 | 05:42:17 | 139, descending |
| Sentinel-1B | 2018-01-09 | 05:42:42 | 139, descending |
| Sentinel-1A | 2018-01-10 | 05:34:45 | 66, descending |
| Sentinel-1A | 2018-01-10 | 05:35:10 | 66, descending |
| Sentinel-1B | 2018-01-11 | 05:26:01 | 168, descending |
| Sentinel-1B | 2018-01-11 | 05:26:26 | 168, descending |
| Sentinel-1A | 2018-01-11 | 17:23:14 | 88, ascending |
| Sentinel-1A | 2018-01-11 | 17:23:39 | 88, ascending |
| Sentinel-1B | 2018-01-12 | 17:14:13 | 15, ascending |
| Sentinel-1B | 2018-01-12 | 17:14:38 | 15, ascending |
| Sentinel-1A | 2018-01-13 | 17:06:47 | 117, ascending |
| Sentinel-1A | 2018-01-13 | 17:07:12 | 117, ascending |

**Figure A2.** Full extent of the RGB composite image TSX 2017-31-12 vs. 2018-01-11 with manually mapped avalanches. *New* avalanches are red, *old* avalanches blue and *unsure* avalanches white. Areas in the radar layover and shadow are masked out (black). TerraSAR-X image orthorectified with the swissALTI3D © 2019 swisstopo (JD100042), reproduced with the authorisation of swisstopo (JA100120).



**Figure A3.** Full extent of the RGB composite image TSX 2018-01-11 vs. 2018-02-02 with manually mapped avalanches. *New* avalanches are red, *old* avalanches blue and *unsure* avalanches white. Areas in the radar layover and shadow are masked out (black). TerraSAR-X image orthorectified with the swissALTI3D © 2019 swisstopo (JD100042), reproduced with the authorisation of swisstopo (JA100120).





*Competing interests.* The authors declare that they have no conflict of interest.

*Acknowledgements.* We thank Elisabeth Hafner from SLF for providing the manual avalanche mask from the optical SPOT-6 data. Lanqing Huang assisted SH to process the S1 data. We are deeply grateful to Irena Hajnsek from ETH Zürich for providing the working environment, the computational resources and her interest for this work. TerraSAR-X data are provided by the German Aerospace Center (Proposal ID: LAN3585). Meteorological data are provided by the Swiss Federal Office of Meteorology and Climatology (MeteoSwiss).






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
