# Peer review of "Snow Avalanche Detection and Mapping in multitemporal and multiorbital Radar Images from TerraSAR-X and Sentinel-1"

_Natural Hazards and Earth System Sciences, 2019_

## Referee Comment (RC1) · Anonymous Referee #1 · 19 Mar 2020

General comments The paper describes various approaches for the systematic mapping of occurred avalanches in order to improve the information provided by avalanche bulletins. It analyzes three different strategies to process radar images (from two different satellites) and cross compares the results. It compares also the mapping results with manual mapping on optical satellite, too. The paper is interesting and well organized. The studied topic is of high interest for the scientific community but also for ski resort manager and local administrator as its application can improve safety management in the area of interest. The only flaw, honestly declared by the authors, is the lack of a real comparison with a ground truth which in the studied case is practically impossible. A minor issue of the dataset is also the revisit time of the analyzed satellites that,

obviously, is not synchronized with main avalanche event but it is widely balanced by the possibility of carrying out an avalanche mapping of entire country like Switzerland (Figure 9 is really impressive when zoomed).

Specific comments The experiment was carried out in a zone featuring a high avalanche activity. It would be interesting, at least at discussion level, to evaluate the performance of the proposed approach in a low frequency area, i.e. to test the capability to detect few sparse events. Figure 2 is really interesting and deserves an improvement for the sake of readability. As the investigated time span is only a fraction of the plotted graph maybe a zoomed plot could be added beside the current one. In the main plot only the AAI and the avalanche type time series should be plotted. In the second, zoomed, one also satellites acquisitions should be added, possibly on a secondary x-axis on top of the plot. Line 134. I guess the topographic relief map is the same used for orthorectification, i.e SwissAlti3D. It would be better to specify it.

Technical corrections In figures 4-8,11, A2, A3 the scale bar is too close to its outline, please increase the distance. In figure 6-8,11, A2, A3 the line fill masks the readability of detected avalanches. Line 18. moasic –>mosaic Line 101. scatters –>scatter Line 367. reduces –>reduced

---

## Referee Comment (RC2) · Markus Eckerstorfer (Referee) · 31 Mar 2020

Dear authors, dear editor! I am very pleased to see this manuscript and I am happy to review it (sorry that it took so long!). In recent years, my group were the only ones, publishing in the field of SAR-borne avalanche detection. We think that the method can assist in reducing uncertainty in public avalanche forecasting and deem the further development of this field very important. I am therefore happy that other groups also show and interest in SAR-borne avalanche detection! This contribution is therefore definitely worth publishing, however, I suggest that the authors consider my comments and questions. I suggest minor revision of the manuscript.

[Figure]

Overall the manuscript is sound and very well written (I cant find any grammatical or spelling mistakes, however, I am not a native speaker). The figures are clear and easy to interpret. The structure of the paper is more or less fine (some comments on it). The biggest problems I see, actually, are that some of the content might not be relevant and the manuscript could potentially be shortened. 1) I do not see much value in presenting detection in single backscatter images and 2) although beautiful to look at, I do not see much value in the multiorbital composite, especially not for automatic detection.

Here are my detailed comments: Figure 1: there is no black rectangle Table 1: could you also indicate orbit number and geometry (asc/desc)? line 53: 55x35 km2, that does not seem correct line 57: where did you download since you had to wait 24 h? line 59: same as line 53. why the square? Figure 2: could you add some more details on the AAI. if I understand it correctly, this is the AAI for entire SUI? from which obser- vations is it calculated (no need to tell us how). what does mixed snow mean (dry high up, wet in the valleys, or due to aspect)? Figure 2: why are the dates of the multiorbital S1 images not shown in Table 1? ok see them in table A1 now! section 2.1: in case you feel like your article is too long, I think this section could be deleted or shortened substantially. you could refer to Yves paper or the SLF special report. line 85: dry slab avalanches at least have three different zones. in case of very wet slab avalanches the zones are more diffuse and in case of loose snow avalanches I would say they are absent. paragraphs 95 - 100: this section reads well but would benefit from some references to microwave scattering in undisturbed snow as well as avalanche debris. paragraph 105: this section reads like discussion. you have not done your analysis yet but conclude already which parts of an avalanche are detectable and why. if you leave this hear, you have to refer to work that tried to assess this at least qualitatively (eg Eckerstorfer et al 2015). line 131: was manual identification done only in TSX data? line 180: why do you use 4 db as a threshold? is that based on literature or on your data? paragraphs 185 - 200: I would consider these paragraphs also as method. please consider moving it there. and for clarification: how do you deal with the fol- lowing situation: dataset A shows 2 separate avalanches which are overlapped by 1

avalanche in dataset B? Figure 6: I am wondering if the readability could be improved by only showing the manually drawn outline, but delete the red lines inside? section 5.3: I am unsure why this exercise of comparing detection in single images and change detection images is of interest? I think it is well established that change detection is the only feasible way to reduce uncertainty in satellite avalanche detection. sections 5.4 and 5.5: this is a very interesting exercise that establishes the upper detection limit of SAR data. could you consider giving some more details here, about how avalanche size plays into detectability? section 5.6: that composite is great, very impressive. just to clarify: you did manual detections in entire Switzerland and found 7361 avalanches? line 290: the POD and FNR are calculated for the red or blue box in your study area? same question also when you compare manual detections. line 290: is the comparison pixel or also feature based? if feature based, how did you handle that for example the automatic detection algorithm split up an avalanche into two features and the manual detection indicates one avalanche? line 335: these statements read confusing. you have mapped an almost similiar number in TSX and Spot-6 images, however, only 68 % and 44 % of the detected avalanches overlap respectively? could you explain this a little bit better please! also, did the same person outline the avalanches in all data? section 6.3: this is a very important section in my view. could you say a little bit more about the size distribution of the avalanches and what the cut-off size is for avalanches not detectable in S1, but clearly visible in TSX. section 6.4: I feel like this is more a repetition of your methods than a discussion of the results. A agree that manual interpretability was improved by all the filtering and smoothing done. however, I somewhat question the use and need of these multiorbital composites, except for visual respresentation of an avalanche cycle. I cannot discern when all these visible avalanches released and which one came first in case of overlapping avalanche activity. this rather long section does not really add much to the overall good discussion of the results. section 6.5: the 4 dB threshold might be probematic and could maybe be replaced with more dynamic thresholding considering backscatter intensity change in individual change detection pairs. section 6.6: I am somewhat confused that you write about

'avalanche differentiation' but I think you are discussing the detectability of avalanches in each of the data!?

---

## Author Response (AR1)

**Author final response**

Dear Editor, Dear Reviewer #1 and Reviewer #2,

thank you very much for spending the time to carefully review the manuscript and for your constructive comments to improving the paper. We carefully revised the paper and considered all points as listed below (equivalent to the final response). Please find also attached a latexdiff document which shows all changes made to the manuscript.

Below are all referee comments (**RC**) by Referee **#1** and Referee **#2** and the corresponding author comments (**AC**). Suggested changes in the manuscript are indicated by *italic* words.

Main changes are

- Removal of the single-image detection which did not provide new insight.

- The essay about meteorological conditions was replaced by a reference.

- A quantitative analysis of the avalanche size distribution was added (see figure).

- The discussion of the multiorbital composites were improved.

The two removals (as suggested by Reviewer **#2**.) shorten the body manuscript which, however, was balanced in length by adding additional information (size distribution, additional discussion). Therefore we will try to shorten the manuscript wherever possible.

**Author response to reviewer #1:**

*General comments*

**RC1:** The paper describes (...). It analyzes (...) and cross compares(...). The paper is interesting and well organized (...) of high interest for the scientific community(...).

**AC1:** Thank you very much for this positive feedback.

**RC1.1:** *cont. general comment*
The only flaw, honestly declared by the authors, is the lack of a real comparison with a ground truth which in the studied case is practically impossible.

**AC1.1:** Yes, the study was based on archive TSX data and such avalanche events can only be predicted very few days in advance. Therefore it was impossible to schedule any ground truth campaigns. We are currently working on a follow-up study which will evaluate if operationally acquired ground based data could be used for ground truth.

**RC1.2:** *cont. general comment*
A minor issue of the dataset is also the revisit time of the analyzed satellites that, obviously, is not synchronized with main avalanche event but it is widely balanced by the possibility of carrying out an avalanche mapping of entire country like Switzerland (Figure 9 is really impressive when zoomed).

**AC1.2:** This might be a misunderstanding but scene selection was indeed not simple. The area of TSX images was defined by available data from the archived data *synchonized* with the two avalanche events. Then, S1 images were *synchonized* by time in incidence angle with the TSX data. The image selection to cover entire Switzerland was defined by the first avalanche event. Unfortunately, TSX data for the second period did not well match the acquisition date of the SPOT-6 images.

To clarify that we did our best to *synchronize* the satellite acquisitions we will modify section 2:
For TSX, no systematic coverage is available *over Switzerland* because TSX acquires data upon request. (...) To cover the two extreme avalanche events around Jan 4th and 22nd 2018 (Fig. 2) with images acquired from identical orbits *as good as possible* we searched the archive for a sequence of images which limited the study area to the Alps of Uri in central Switzerland. *The orbit repeat time defines the dates and limits the revisit time to 11 d for the first event and 22 d for the second one (one missing acquisition).* The Sentinel-1 images were carefully selected to cover the first main avalanche event. We suggest to add to section 2: *To study the possibility of detecting avalanches for entire Switzerland for the first avalanche event, S1 acquisitions were carefully selected from multiple orbits during a 5 day period with from before and after the first event (Table A1).*

*Specific comments*

**RC2:** The experiment was carried out in a zone featuring a high avalanche activity. It would be interesting, at least at discussion level, to evaluate the performance of the proposed approach in a low frequency area, i.e. to test the capability to detect few sparse events.

**AC2:** The first criterion to detect avalanches is a brightness threshold of 4 dB which is better fulfilled as long as no new avalanches overlap with old ones. That means that sparse events are more likely to detected than overlapping avalanches in densely avalanche-covered areas. A later criterion is that avalanches must have a certain size to be detected which efficiently filters out noise in areas with sparse events. To address these two point, we suggest to add after the first sentence in section 6.5 (discussion: automated avalanche detection):
For both, TSX and S1 images the implemented avalanche detection algorithm performs with reasonable results, at least when the number of overlapping avalanches is low. *That means that a few sparse events are more likely to be detected than overlapping clusters of avalanches.*

**RC3:** Figure 2 is really interesting and deserves an improvement for the sake of readability. As the investigated time span

is only a fraction of the plotted graph. Maybe a zoomed plot could be added beside the current one. In the main plot only the AAI and the avalanche type time series should be plotted. In the second, zoomed, one also satellites acquisitions should be added, possibly on a secondary x-axis on top of the plot.

**AC3:** Thanks for this suggestion. Honestly, I spent multiple days working on this figure and I tried already (before submission) to improve the figure in the direction suggested by you. However, I was not satisfied with the solution main+subplot because to show the intensity of the two extreme events on 2018-01-04 and 2018-01-22 the y-axis of the graph should not be cut more than it already is (AAI = 1200). Zooming to the study period late December - early February leaves no space for the legend and y-axis label; furthermore, a zoomed sub-graph would require a lot of vertical space when not cutting of the y-axis. I think the readability will *appear* improved in the two-column format (without changing the size) because then it fills the full width of one of the two columns. I agree, that in the 1-column manuscript, where it is displayed with half page width, it appears a bit small.

**RC4:** Line 134. I guess the topographic relief map is the same used for orthorectification, i.e SwissAlti3D. It would be better to specify it.

**AC4:** Yes, we used the Alti3D for that evaluating the flow path in single-image avalanche detection. However, we like to follow the suggestion of Reviewer **#2** to remove the results and discussion about single-image detection. According to Reviewer **#2**, there is no point to evaluate a technique which is behind the state-of-the-art of two-image change detection.

*Technical corrections*

**RC5:** Technical corrections In figures 4-8,11, A2, A3 the scale bar is too close to its outline, please increase the distance.

**AC5:** done.

**RC6:** In figure 6-8,11, A2, A3 the line fill masks the readability of detected avalanches.

**AC6:** The idea of these images is to show and compare the masks. - Figure 6 will be removed as suggested by Reviewer #2.
- The sole purpose of Figure 7 is to compare masks. Even when the line fill masks would be removed and only an outline would be used to show the avalanche area, the outline would either bias the reader or mask the avalanche edge (for the optical image 7).
- For Fig. 8 we will add to the caption that *the S1 image in the background is shown without mask in Fig. 4c*.
- The same holds for Fig. 11 (TSX) which is already shown in Fig. 4b without mask.
- In Figure A2 and A3, the relatively large spaces between

the line fill mask make it possible to see the backscatter image when zooming in.

**RC7:** (typos)
Line 18. moasic –>mosaic
Line 101. scatters –>scatter
Line 367. reduces –>reduced

**AC7:** Thanks; all typos have been corrected.

**Author response to reviewer #2:**

*General comments*

**RC8:** Overall the manuscript is sound and very well written (...). The biggest problems I see, actually, are that some of the content might not be relevant and the manuscript could potentially be shortened.

**AC8:** We will consider your suggestions and will remove the single-avalanche detection (reduces the manuscript length by 2 pages and 2 figures) and will replace the description of meteorological event (1/2 page) by a reference. (See RC8.1/24 and RC16).

**RC8.1:** I do not see much value in presenting detection in single backscatter images.

**AC8.1:** See **AC8** and **RC24**. We will remove the single backscatter analysis. This will shorten the manuscript by almost 2 pages and will also remove 2 figures.

**RC8.2:** although beautiful to look at, I do not see much value in the multiorbital composite, especially not for automatic detection.

**AC8.2:** This contradicts a bit RC26 "very impressive", doesn't it? We like explicitly point out here the three main advantages of multiorbital composites:

    + the reduction of invisible areas (mainly layover).

    + the enhancement of resolution for near-layover areas.

    + the reduction of radar speckle noise.

However, there are also some apparent disadvantages of the multiorbital composites which we will discuss in the manuscript (see also RC31):

    - *loosing some temporal resolution (when the time difference between asc+desc acquisitions is large). Under unfortunate cases, when avalanches occur between the averaged acquisitions, they appear half as bright and are more likely to be missed. However, the less time elapses between asc and desc acquisitions the more likely it is that avalanches are captures with both orbits which improves the SNR for detection. In our study, acquisitions were carefully selected to best image the avalanches of the first avalanche event on 2018-01-04. We will add this information to the discussion of the multiorbit composites and will mention also in the conclusion that especially the ratio between the elapsed time between asc and desc acquisition and the revisit-time must be minimized.*

We think the above disadvantage is only *apparent*, because we draw important conclusions from the multiorbit averaging which hint in the direction that local resolution weighting (Small, 2012) will further improve our results (and reduces the effect of loosing temporal resolution) we will add to the discussion: *In mountaineous regions, LRW applies already unequal weights for ascending and descending acquisitions which further decreases the probability that avalanche visibility is reduced by the multiorbital averaging.*

While reconsidering the advantage of the multiorbital composites we also realized that *shadow areas are not improved because they are located in layover when imaged from the opposite orbit.*

*Specific comments*

**RC9:** Figure 1: there is no black rectangle.

**AC9:** It's in the inset. We will modify the figure caption to: *The black rectangle in the insets shows the full footprint of the TSX scene over Switzerland.*

**RC10:** Table 1: could you also indicate orbit number and geometry (asc/desc)?

**AC10:** sure.

**RC11:** line 53: 55x35 km2, that does not seem correct.

**AC11:** This number was measured from the scene extent. It is close to the standard footprint size of TSX in single-pol stripmap mode. We will hint to the *inset of Fig. 1* which shows the size of the footprint to avoid misunderstanding.

See also **RC13** "Why the square?": When specifying an area I think the correct unit should be $km^2$. It might seem a bit awkward to write $55 \times 35\ km^2$ but in my opinion it's the best way to specify the extent. Alternative formulations would be $55\ km \times 35\ km$ or $1925\ km^2$ (I have not found any standards for areas in NHESS).

**RC12:** line 57: where did you download since you had to wait 24 h?

**AC12:** This might be a misunderstanding: for this study we did not analyze any time-critical data and did not had to wait. The 24 h describe the general availability of S1 data. As the current reference does not contain this information, we will update the reference where this information is provided: "Global products will be systematically generated for all acquired data. (...) These products are made available (...) in any case within 24 h after observation." (ESA, 2012, p.35). See also https://sentinel.esa.int/web/sentinel/missions/sentinel-1/data-distribution-schedule

**RC13:** line 59: same as line 53. why the square?

**AC13:** See **AC11**.

**RC14:** Figure 2: could you add some more details on the AAI. if I understand it correctly, this is the AAI for entire SUI? From which observations is it calculated (no need to tell us how).

**AC14:** I will add to the caption: *The avalanche activity index is the weighted sum of all reported avalanches for Switzerland (Schweizer et al., 2003, 1998).* The AAI depends on visibility, because avalanches can only be *reported* when visible for a human observer.

**RC14.1:** What does mixed snow mean (dry high up, wet in the valleys, or due to aspect)?

**AC14.1:** We'll add to the caption: *"mixed snow" indicates dry snow avalanches which started high up but were slowed down at medium altitude by wet snow.*

**RC15:** Figure 2: why are the dates of the multiorbital S1 images not shown in Table 1? ok see them in table A1 now!

**AC15:** We will add the reference to table A1 to the caption of table 1 and Fig. 2.

**RC16:** section 2.1: in case you feel like your article is too long, I think this section could be deleted or shortened substantially. You could refer to Yves paper or the SLF special report.

**AC16:** We'll delete this section and refer in Sect. 2 to Yves paper (Bühler et al., 2019) instead. We do anyway not refer to this section in the remainder of the paper.

**RC17:** line 85: dry slab avalanches at least have three different zones. In case of very wet slab avalanches the zones are more diffuse and in case of loose snow avalanches I would say they are absent.

**AC17:** Thanks for noting this. We will reformulation the sentence to *Fig. 3 illustrates a classification scheme from (International Commission of Snow and Ice, 1981). The scheme suggests that all types of snow avalanches can be composed into three different zones, however, for some avalanche types (e.g. loose snow avalanches) zones can be difficult to differentiate.*

**RC18:** paragraphs 95 - 100: this section reads well but would benefit from some references to microwave scattering in undisturbed snow as well as avalanche debris.

**AC18:** There is a large number of publications of various quality study the interaction of microwaves with snow under different conditions. Instead of picking publications describing specific models for microwave scattering at rough surfaces, we think the best approach is similar to (Eckerstorfer and Malnes, 2015) and to provide a general, qualitative description of scattering physics: *Currently, there exists no specific model tailored to the backscatter properties of snow avalanches (cf. (Eckerstorfer and Malnes, 2015, Sect. 5.3)), however, general scattering physics from bi-continuous media and rough surfaces can be applied. In that sense, scattering in snow increases with the spatial correlation length of ice grains (Wiesmann et al., 1998) and also with increased surface- and interface roughness and with decreasing incidence angle $\theta$ (Leader, 1971; Fung and Eom, 1982; Kendra et al., 1998).*

**RC19:** paragraph 105: this section reads like discussion. You have not done your analysis yet but conclude already which parts of an avalanche are detectable and why. if you leave this here, you have to refer to work that tried to assess this at least qualitatively (eg Eckerstorfer et al 2015).

**AC19:** We agree (See also comment above). Basically, we encounter the same problem as stated in (Eckerstorfer and Malnes, 2015): "To date, no appropriate electromagnetic backscatter model for disturbed snowpacks like avalanche debris exists. Due to this lack of an appropriate model (...) we cannot give an exact theoretical or statistical explanation.". We agree and I think that such an appropriate model is neither required nor of particular help because for avalanches the general scattering physics of rough surfaces should apply and explain already qualitatively the observed scattering behavoir. Still, as we follow similar ideas it makes a lot of sense to cite (Eckerstorfer and Malnes, 2015, Sect 5.3).

**RC20:** line 131: was manual identification done only in TSX data?

**AC20:** We will add to section 4.3 *From the images (TSX and S1) avalanche outlines were drawn manually.* (Note, that the single-image section you are referring to will be removed).

**RC21:** line 180: why do you use 4 db as a threshold? is that based on literature or on your data?

**AC21:** We will add: *The threshold was determined empirically based on TSX data but other authors also used thresholds of 4–6 dB (Eckerstorfer et al., 2019; Karbou et al., 2018; Vickers et al., 2016).*

**RC22:** paragraphs 185 - 200: I would consider these paragraphs also as method. please consider moving it there.

**AC22:** We will add a method-subsection: *"comparison between mapping results".*

**RC22.1:** and for clarification: how do you deal with the following situation: dataset A shows 2 separate avalanches which are overlapped by 1 avalanche in dataset B?

**RC22.1:** The problem over overlapping avalanches will be detailed in the above mentioned subsection. The problem is also already discussed in the discussion section 6.6 "avalanche differentiation".

**RC23:** Figure 6: I am wondering if the readability could be improved by only showing the manually drawn outline, but delete the red lines inside?

**AC23:** In general, see **AC6** (figures without hashed lines are shown in Fig. 4). This specific figure will be removed anyway (see also **RC8.1** and **RC24**).

**RC24:** section 5.3: I am unsure why this exercise of comparing detection in single images and change detection images is of interest? I think it is well established that change detection is the only feasible way to reduce uncertainty in satellite avalanche detection.

**AC24:** See general comment **RC8.1** (will be removed). Instead we provide a few references at the begin of the method section 4.3 and reasons why change detection is the preferred way: *Although well visible avalanches could be manually detected in single radar images, single images are difficult to analyze with automatic methods. As radar systems carry their own illumination system the backscatter signal is primarily determined by topography and land cover type. It is therefore common practise to analyze temporal backscatter variations to separate sudden changes from signals of stable topographic and land cover features (Wiesmann et al., 2001; Eckerstorfer and Malnes, 2015).*

**RC25:** sections 5.4 and 5.5: this is a very interesting exercise that establishes the upper detection limit of SAR data. Could you consider giving some more details here, about how avalanche size plays into detectability?

**AC25:** Thanks for this comment. Though lengthening the manuscript, *we will add a new section (5.8) presenting the systematic analysis of detected avalanche sizes.* We found this analysis very interesting and important. See attached figure.

We will add to Sect. 5.4: *We did not found significant differences in area for the lower detection limits: for both, TSX and SPOT-6 the smallest detectable avalanches had an area of $500\,m^2$. (reference to new Sect. 5.8).*

We will add to Sect. 5.5: *We found that the smallest avalanches detected by S1 have an area of around $2000\,m^2$ (reference to new Sect. 5.8).*

**RC26:** section 5.6: that composite is great, very impressive. just to clarify: you did manual detections in entire Switzerland and found 7361 avalanches?

**AC26:** Yes! But I only counted them systematically using screen-fitting boxes of 12x12 km (1200x1200 px) which was pretty fast (a bit more than 4 hours). We will add: we *manually* counted 7361 avalanches *but did not draw any polygons.*

**RC27:** line 290: the POD and FNR are calculated for the red or blue box in your study area? same question also when you compare manual detections.

**AC27:** POD and FNR were calculated for *the entire study area (red polygon in Fig. 1).*, not only for the blue visualization window. We will add this information to figure captions and the text.

**RC28:** line 290: is the comparison pixel or also feature based? if feature based, how did you handle that for example the automatic detection algorithm split up an avalanche

[Figure]

[Figure]

**Figure 1.** (a): Classification of avalanche area into size classes. (b): the cumulative avalanche area plotted over avalanche size reveals that the smallest avalanches detected by TSX and SPOT-6 is about $500\,m^2$ and $2000\,m^2$ for S1. It may surprise that in the study region the total avalanche area of SPOT-6 is an order of magnitude(!) larger than the total area of radar-detected *new* avalanches. Still, a factor of three remains when comparing the area of all (*new, old, unsure*) radar detected avalanches with SPOT-6 (no age classification, only 24% of outlines clearly visible; 76% of avalanches outlines were estimated from partially visible avalanche patches (Bühler et al., 2019)). Considering that with radar mainly the deposition zone has been detected and mapped this difference is reasonable.

into two features and the manual detection indicates one avalanche?

**AC28:** The comparison is feature based. See first result section (line 185-196). This section will be moved to the method (**RC22**). See also discussion 6.6 and **RC22**.

**RC29:** line 335: these statements read confusing. you have mapped an almost similar number in TSX and Spot-6 images, however, only 68% and 44% of the detected avalanches overlap respectively? Could you explain this a little bit better please!

Also, did the same person outline the avalanches in all data?

**AC29:** See also **RC22, RC28**. We suggest to add to the results section "SPOT-6 vs. TSX comparison": *Avalanche were mapped independently by E. Hafner in SPOT-6 images (Bühler et al., 2019) and by the second author of this work in TSX*

*images. The polygons differed significantly, however, most likely because different features (avalanche origin, path, deposition zone) are visible in optical and radar images. Therefore we decided for a feature-based comparison, i.e. overlapping polygon are considered as avalanches detected in both data sets. Avalanches split up into discontiguous polygons were counted separately, also if all polygons overlap with one single large polygon in the other data set (see also new Sect. 4.7 ->**RC22**).* We hope, that explaining the reason of split up avalanches solves this confusion. The difference 68% vs. 44% is also explained by the differentiability of adjacent avalanches in the succeeding paragraph (339-342).

**RC30:** section 6.3: this is a very important section in my view. could you say a little bit more about the size distribution of the avalanches and what the cut-off size is for avalanches not detectable in S1, but clearly visible in TSX.

**AC30:** Based on the added size distribution we suggest to add (in the sense of **RC25**): *The size of smallest detectable avalanches for TSX are "medium" avalanches ($500 - 10\,000\,m^2$) with a width of more than $20\,m$. S1 misses mainly "medium" avalanches smaller than $2\,000\,m^2$. Similar results for S1 with a minimum cutoff of $4\,000\,m^2$ were found by Eckerstorfer et al. (2019).*

**RC31:** section 6.4: I feel like this is more a repetition of your methods than a discussion of the results. I agree that manual interpretability was improved by all the filtering and smoothing done. however, I somewhat question the use and need of these multiorbital composites, except for visual representation of an avalanche cycle. I cannot discern when all these visible avalanches released and which one came first in case of overlapping avalanche activity. this rather long section does not really add much to the overall good discussion of the results.

**AC31:** We will carefully check this section to decide what should be shortened (*NL mean discussion*) or moved to the method part (*some resolution discussion will be (re)moved to the method section*). But we think especially the multiorbit-composites requires a detailed discussion as it contains may promising approaches and outlooks. See also **RC8.2**.

For discern which avalanche came first, see suggested changes in **RC8.2**.

**RC32:** section 6.5: the 4 dB threshold might be probematic and could maybe be replaced with more dynamic thresholding considering backscatter intensity change in individual change detection pairs.

**AC32:** We will add this suggestions. *A dynamic threshold based on backscatter changes in individual image pairs could improve these results (Eckerstorfer et al., 2019).*

**RC33:** section 6.6: I am somewhat confused that you write about 'avalanche differentiation' but I think you are discussing the detectability of avalanches in each of the data!?

**AC33:** Detectability and differentiation differ in the following sense: one method could better detect weakly visible large avalanches than another method. or: one method could better differentiate a large patch into multiple smaller avalanches. We mean the latter: *
[revised manuscript text omitted]
 3–, the avalanche danger level was generally 3 (considerable) but raised to 4 (high) on Jan 4/, and on the for a major part of the Swiss alps (SLF, 2018a). During sunny days from Jan 13–the snow line raised up to followed by almost daily precipitation and strong winds from north-west

[Figure]

**Figure 2.**  The avalanche activity index is the weighted sum of all reported avalanches for Switzerland (Schweizer et al., 2003, 1998). Dry snow avalanches which started high up but were slowed down at medium altitude by wet snow are indicated as "mixed snow" in the legend. Satellite acquisitions dates are indicated by arrows. Images for the multiorbital S1 composite were acquired during the gray shaded periods (see also Table A1). Figure modified after Winkler et al. (2019).

on Jan 16–. Wind speeds of over occurred during the storms "Evi" and "Friedericke" on Jan and . On Jan 20/extreme snowfall was registered. From Jan 21–, the avalanche activity reached a new three-day record for the past 19 years since the avalanche winter in 1999 (Winkler et al., 2019). Due to the extraordinary avalanche situation, the avalanche danger level was raised to 5 (very high) on Jan 21/(SLF, 2018c). After Jan the situation eased and temperatures were very warm with a snowline rising from to over until end of January. The avalanche activity was low and the avalanche danger level was mainly moderate for the analyzed area. On Feb temperature dropped and around of snow fell (SLF, 2018d). Therefore, snow conditions were slightly different between the TSX and SPOT-6 data but only a few new avalanches occurred (Fig. 2).

**3 Radar backscatter physics of avalanches**

[revised manuscript text omitted]

$$\sigma^0_{snow}(\theta) = \sigma^0_{surf}(\theta) + \sigma^0_{vol}(\theta) + \sigma^0_{ground}(\theta) + \sigma^0_{inter.}(\theta) \qquad (1)$$

**4 Methods**

**4.1 Data preprocessing**

All radar products were downloaded in the single look complex (SLC) format. The data were preprocessed with the ESA SNAP Sentinel-1 toolbox and also with the GAMMA software for comparison. The workflow using GAMMA was implemented with Nextflow (Di Tommaso et al., 2017) to speed up execution and code development and to ensure a reproducible analysis. Preprocessing consists of coregistration, multilooking for reduction of radar speckle (TSX: 6×5 px, S1: 4×1 px), orthorectification, and generation of radar shadow and layover masks. The SNAP workflow for S1 images is shown in Fig. A1. We did not apply any radiometric terrain correction as the visible topography helps to identify the avalanche path direction.

For orthorectification we used the Swiss elevation model SwissAlti3D (2013) downsampled from 2 m to 30 m resolution. We noticed, however, that despite of using the same DEM and output resolution, sharp topographic features seem to be better orthorectified with the GAMMA software which might use a more precise spatial interpolation. The radar images were orthorectified to a resolution of 5x5 m (TSX) and 15x15 m (S1) and the backscatter signal in dB was saved to geotiff files. The exact radiometric normalization is irrelevant, because we did not apply any radiometric terrain correction (Small, 2011) and different ellipsoidal corrections ($\sigma_E^0, \gamma_E^0$) differ only by almost constant factors. Since the TSX data was acquired with a single polarization (the co-polar channel HH) we also used only the co-polar channel (VV) of the two available polarizations of S1 to obtain a fair comparison. For the multiorbital composites, we used both polarizations of S1 (VV, VH).

**4.2**

~~For avalanche detection by visual inspection in single images, areas with radar shadow and layover were masked out. The images (as shown in Fig. ??) were then systematically searched for bright features matching the tongue shaped geometry of the avalanche deposition zone. Potential avalanches were manually contoured to create an avalanche mask. For uncertain cases, a topographic relief map was used to decide if the identified shape corresponds to a possible flow path.~~

**4.2 Two-image composite avalanche detection**

Although avalanches could be manually detected in single radar images they are difficult to analyze with automatic methods. As radar systems carry their own illumination system the backscatter signal is primarily determined by topography and land cover type. It is therefore common practise to analyze change detection images to separate sudden backscatter changes from stable topographic and land cover features (Wiesmann et al., 2001; Eckerstorfer and Malnes, 2015). To correct for large-scale backscatter changes due to wet snow a 500 m highpass filter was applied to the backscatter difference between  two consecutive images. Examples for TSX and S1 are shown in Figs. 4a and 4b. To create the images, the backscatter intensities in dB were normalized by clipping the lower and upper 1%. Consecutive images were then stored in the channels [R, G, B] = [img2, img1, img1] so that backscatter changes are well visible by the red-cyan contrast. From the images (TSX and S1) avalanche outlines were drawn manually.

The  change detection images  allow for a temporal classification of avalanches into three classes  (new, old, unsure). *New* avalanches appear red because of increased backscattering and are therefore assumed to have occurred between the first and the second acquisition. *Old* avalanches, with a decreasing backscatter signal, appear blue are therefore assumed to have occurred before the first acquisition. Bright features with  unchanged backscatter intensity appear almost white and are classified as *unsure* if they look like avalanches.

**4.3 Multiorbital composite image for Switzerland**

 The free and systematic availability of S1 radar images and the short revisit period of  six days allow for creation of an RGB composite change detection image covering entire Switzerland. Therefore, 12 images, acquired between 2017-12-28 and 2018-01-01 from different orbits, were combined into an image before the first avalanche event (Jan 4th). Another 12 images, acquired between 2018-01-09 and 2018-01-12 with an identical imaging geometry, were used for the post-avalanche event image. The images (listed in Table A1) were preprocessed according to Sect. 4.1. To reduce radar speckle we averaged both polarizations and weighted the cross-pol channel (VH) by the ratio $a$ of the co- and cross-pol backscatter intensities averaged over the acquisition footprint:

$$S = \frac{S_{VV} + a\,S_{VH}}{1 + a} \qquad \text{with} \qquad a = \frac{\langle S_{VV} \rangle}{\langle S_{VH} \rangle} \qquad (2)$$

Then, the weighted mean was converted to dB and scenes from different ascending and descending orbits were aver-

Subset of single radar image TSX(01-11).

[Figure]

(a) Subset of change detection image tsx(12-31/01-11).

(b) Subset of change detection image S1(12-31/01-12).

(c) Subset of S1 multiorbital change detection image from the two data sets 2017-12-28 – 2018-01-01 vs. 2018-01-09 – 2018-01-12.

(d) Subsets of the study area: Multiorbital S1 change detection image as in (ac) single radar image, but with non-local mean filter applied.

**Figure 4.** (ba,eb) TSX and S1 change detection images , and of a subsets of the study area (dcf. Fig. 1)
[revised manuscript text omitted]